# Should I Stop or Should I Go:
# Early Stopping with Heterogeneous Populations

**Hammaad Adam**[*]
Massachusetts Institute of Technology

**Fan Yin**[†]
Microsoft Corporation

**Huibin (Mary) Hu**
Microsoft Corporation

**Neil Tenenholtz**
Microsoft Research

**Lorin Crawford**
Microsoft Research

**Lester Mackey**
Microsoft Research

**Allison Koenecke**
Cornell University

## Abstract

Randomized experiments often need to be stopped prematurely due to the treatment having an unintended harmful effect. Existing methods that determine when to stop an experiment early are typically applied to the data in aggregate and do not account for treatment effect heterogeneity. In this paper, we study the early stopping of experiments for harm on heterogeneous populations. We first establish that current methods often fail to stop experiments when the treatment harms a minority group of participants. We then use causal machine learning to develop CLASH, the first broadly-applicable method for heterogeneous early stopping. We demonstrate CLASH's performance on simulated and real data and show that it yields effective early stopping for both clinical trials and A/B tests.

## 1 Introduction

Randomized experiments are the gold-standard method of determining causal effects, whether in clinical trials to evaluate medical treatments or in A/B tests to evaluate online product offerings. The sample size and duration of such experiments are typically specified in advance. However, there are often strong ethical and financial reasons to stop an experiment before its scheduled end, especially if the treatment shows early evidence of harm [7]. For example, if early data from a clinical trial demonstrates that the treatment increases the rate of serious side effects, it may be necessary to stop the trial to protect experimental participants [9].

A variety of statistical methods can be used to determine when to stop an experiment for harm [22, 39, 14, 13]. Investigators in both clinical trials and A/B tests will often choose to use a subset of these methods—collectively referred to as "stopping tests"—based on the specifics of their domain. Stopping tests not only identify harmful effects from early data, but also limit the probability of stopping early when the treatment is not harmful. However, stopping tests are typically applied to the data in aggregate (i.e., "homogeneously") and do not account for heterogeneous populations. For example, a drug may be safe for younger patients but harm patients over the age of 65. While a growing body of literature has studied how to infer such heterogeneous effects [see, e.g., 41], little prior research has described how to adapt stopping tests to respond to heterogeneity.

We continue the above example to illustrate why stopping tests should account for heterogeneity. Consider a clinical trial for a drug such as warfarin, which has no harmful effect on the majority of the population but increases the rate of adverse effects in elderly patients [32]. Using a simple simulation, we demonstrate that if elderly patients comprise 10% of the trial population, then applying a stopping

---

[*]Corresponding author (hadam@mit.edu).

[†]Current affiliation: Amazon. Work performed while at Microsoft (prior to joining Amazon).

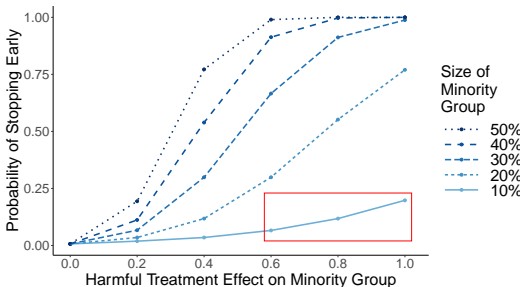

Figure 1: Stopping tests applied homogeneously may not stop trials in which the treatment harms only a minority group of participants. For example, if the treatment harms 10% of all participants, the trial will stop less than 20% of the time, even if the harmful treatment effect is strong (boxed in red). We plot the stopping probability of a commonly used stopping test (O'Brien-Fleming) in situations where the treatment harms the minority group but has no effect on the majority group.

test homogeneously would rarely stop the trial for harm (Fig. 1). In most cases, the trial continues to recruit elderly patients until its scheduled end, many of whom will be harmed by their participation. This outcome violates the bioethical principle of non-maleficence [37] and is clearly undesirable.

In this paper, we consider early stopping for harm with heterogeneous populations. We first formalize the problem of heterogeneous stopping and establish the shortcomings of the prevailing homogeneous approach. We then present our novel method: Causal Latent Analysis for Stopping Heterogeneously (CLASH). CLASH uses causal machine learning to infer the probability that a participant is harmed by the treatment, then adapts existing stopping tests to better detect this harm. CLASH allows an investigator to use their stopping test of choice: it is thus flexible and easy-to-use. We theoretically establish that, for sufficiently large samples, CLASH is more likely to stop a trial than the homogeneous approach if the treatment harms only a subset of trial participants. CLASH also does not stop trials unless a group is harmed; it thus increases statistical power while limiting Type I error rate. We demonstrate these properties in an extensive series of experiments on synthetic and real data, and establish that CLASH leads to effective heterogeneous stopping across a wide range of domains.

While a small number of papers have attempted to address the problem of heterogeneous early stopping, they have relied on restrictive assumptions, limiting their applicability. Thall and Wathen [35] require knowledge of the source of heterogeneity, which is rarely available in practice. Yu et al. [42] allow for unknown groups but only model linear heterogeneity. Yu et al. [43] relax the linearity restriction but do not handle time-to-event data commonly observed in clinical trials. In contrast, CLASH does not require prior knowledge, makes no parametric assumptions, and works with any data distribution. It is thus the first broadly applicable tool for early stopping with any number of heterogeneous groups.

We emphasize that early stopping is a nuanced decision. For example, if a treatment harms only a subset of participants, it may be desirable to stop the experiment only on the harmed group but continue it on the rest of the population. In other situations, it may make sense to stop the trial altogether. The decision to stop a trial to protect one group may disadvantage another group that would have benefited from the treatment; any decision on early stopping must thus carefully weigh the treatment's potential benefit, the size of the harmed group, the nature of the harm, and other ethical considerations. Our method is not intended to replace such discussion. It can, however, serve as an aid for investigators to make difficult decisions on early stopping.

## 2 Background and Setup

### 2.1 Randomized Experiment

Consider an experiment that evaluates the effect of a binary treatment $D \in \{0, 1\}$ on an observed outcome $Y$. $D$ is assigned uniformly at random to participants in the experiment. $Y$ is chosen specifically to measure harm and may be different from the experiment's primary outcome of interest. For example, in a clinical trial, $Y$ could convey a primary outcome such as mortality or a secondary outcome such as the occurrence of serious side effects. We allow for both scenarios, as experiments

often care about harm on multiple dimensions. Assume that an increase in $Y$ is harmful. The harm caused by the treatment can be measured by the average treatment effect (ATE) of $D$ on $Y$, ATE $= \mathbb{E}[Y|D=1] - \mathbb{E}[Y|D=0]$. A positive ATE implies that the treatment is harmful, while a negative ATE implies that it is beneficial. Note that the ATE has a natural estimator: the difference in the observed means of $Y$ between the treated and untreated groups.

Now, it is not necessary that every individual in the population responds to the treatment in the same way. The treatment may harm some individuals, benefit some, and have no effect on others. Assume that the population can be divided into $M$ discrete groups, where random variable $G$ conveys group membership. For group $g$, we define the conditional average treatment effect (CATE) $\tau(g) = \mathbb{E}[Y|D=1, G=g] - \mathbb{E}[Y|D=0, G=g]$. Note that in general, we do not observe $G$. Rather, we typically observe a set of covariates $X$ that correlate with group membership. We define the CATE for a participant with covariates $x$ as $\tau(x) = \mathbb{E}_{g|x}[\tau(g)]$.

The experiment can recruit a maximum of $N$ participants and is run for $T$ time steps, where the values of $N$ and $T$ are set at the experiment's start. For simplicity, assume that we recruit two participants at each time step—one in treatment and one in control—and so $T = N/2$. For each participant $i$, we measure treatment assignment $d_i$, covariates $x_i$, and observed outcome $y_i$, but do not observe group membership $g_i$. The expected effect of the treatment on participant $i$ is given by $\tau(g_i)$. We assume that $(d_1, x_1, y_1, g_1), \ldots, (d_N, x_N, y_N, g_N)$ are independent and identically distributed.

## 2.2 Early Stopping for Harm

If the treatment is harmful, it is often important to end the experiment as early as possible, to minimize harm for participants and financial loss for the experimenter. Clinical trials and A/B tests usually build in interim checkpoints, where the data collected thus far is evaluated to make a decision on whether to continue the experiment. However, it is also important to *not* stop the experiment at these checkpoints unless the treatment is actually harmful; incorrectly stopping early could result in lost innovation or societal benefit. Achieving this balance requires using dedicated statistical methods for early stopping, collectively known as "stopping tests."

A large body of literature has studied stopping tests. Frequentist methods like the Pocock adjustment [23], O'Brien-Fleming (OF) adjustment [22], and alpha-spending [10] are common in clinical trials with a pre-specified schedule of checkpoints. True sequential methods that allow for continuous monitoring are often based around Wald's Sequential Probability Ratio Test (SPRT) [39], such as the 2-SPRT [19], MaxSPRT [16], and mixture-SPRT (mSPRT) [26, 14]. Other sequential methods are based on martingales [3, 25] and testing by betting [40]. Such tests are more practical in A/B testing scenarios, where investigators can monitor responses as they arrive. There is also a growing interest in Bayesian designs and stopping tests, both for clinical trials [4] and A/B tests [33]. App. A summarizes some commonly-used stopping tests.

While these stopping tests differ in their details, many have the same general form: compute a test statistic from the hitherto observed data $\lambda_n(y_{1:n}, d_{1:n})$, then stop the experiment if this test statistic exceeds some bound $b(\alpha)$. Here, $n$ is the number of outcomes observed thus far and $\alpha$ is the desired bound on the rate of unnecessary stopping (i.e., stopping when the treatment is not harmful). For clarity, we provide a concrete example of a stopping test: the OF-adjusted z-test. Assume that $Y \sim \mathcal{N}(D\mu, \sigma^2)$, where $\sigma^2$ is known but $\mu$ is not. The treatment effect is homogeneous, and so the ATE (i.e., $\mu$) conveys if the treatment is harmful. To test between a null hypothesis of no harmful effect ($H_0 : \text{ATE} \leq 0$) and an alternate hypothesis of a harmful effect ($H_1 : \text{ATE} > 0$), we use

$$\lambda_n^{OF} = \frac{\sqrt{n}}{\sqrt{2\sigma^2}} \left( \frac{\sum_{i=1}^n y_i d_i}{n/2} - \frac{\sum_{i=1}^n y_i (1-d_i)}{n/2} \right). \tag{1}$$

The OF-adjusted z-test compares this test statistic to an appropriate bound $b^{OF}(\alpha)$, which depends both on the number and timing of checkpoints. For example, if we were to conduct one checkpoint halfway through the trial, then $b^{OF}(0.05) = 2.37$. If $\lambda_{N/2}^{OF} > b^{OF}$, the test stops the experiment for harm; else, the experiment continues. Using the OF adjustment guarantees that if the treatment is not harmful (i.e., ATE $\leq 0$), then the probability of stopping the experiment is at most $5\%$.

## 2.3 Stopping with Heterogeneous Populations

Stopping tests are typically applied to the data in aggregate and define harm in terms of the ATE. While this approach is reasonable if the treatment effect is homogeneous, there is no guarantee that it

will stop the experiment if the treatment only harms a subset of participants. For example, consider a situation in which there are two equally sized groups with equal but opposite treatment effects, that is, $p(G = 0) = p(G = 1)$ and $\tau(0) = -\tau(1)$. The ATE with this mixture distribution is zero, and so any stopping test with $H_0$ : ATE $\leq 0$ is designed to continue to completion at least $(1 - \alpha)\%$ of the time. Unfortunately, the failure to stop means that half of the trial participants will be harmed.

Experiments with heterogeneous populations thus require a more precise definition of harm. To comply with the bioethical principle of non-maleficence, experiments should be stopped not only if the treatment is harmful in aggregate (i.e., ATE $> 0$), but also if the treatment harms any group of participants (i.e., $\exists\, g$ s.t. $\tau(g) > 0$). We define the null hypothesis of no group harm,

$$H_0 : \tau(g) \leq 0 \ \forall g \in \{1, ..., M\}. \tag{2}$$

An effective stopping test must fulfill two conditions. First, it must have high stopping probability if the treatment harms any group of participants. Second, it must limit the probability of stopping if no group of participants is harmed (i.e., limit the "rate of unnecessary stopping"). Note that these conditions are equivalent to high statistical power and low Type I error rate relative to (2).[3] For the rest of the paper, we use these two metrics—the stopping probability if any group is harmed and the rate of unnecessary stopping—to evaluate stopping tests with heterogeneous populations.

Fig. 1 demonstrates the limitations of applying stopping tests homogeneously to heterogeneous populations. We simulate an experiment that runs for 1,000 time steps and recruits 2,000 participants. Participants come from two groups, with $G \in \{0, 1\}$ indicating membership in a minority group. The treatment has no effect on the majority group but harms the minority with treatment effect $\theta$, with $Y \sim \mathcal{N}(\theta \times D \times G, \sigma^2)$ and $\sigma^2 = 1$. The trial has one checkpoint at the halfway stage and uses an OF-adjusted z-test to decide whether to stop for harm. We vary $\theta$ from 0 to 1 and $\mathbb{P}(G = 1)$ from 0.1 to 0.5, and plot the stopping probability across 1,000 replications. We find that as the size of the harmed group decreases, stopping probability falls significantly. Continuing our warfarin example from Sec. 1, if elderly patients comprise 10% of trial participants (lowest line in Fig. 1), then the stopping probability is less than 20%, even if the treatment has a large harmful effect ($\theta = 1$).

## 3 Causal Latent Analysis for Stopping Heterogeneously (CLASH)

Thus far, we have introduced early stopping for harm and established that existing methods applied homogeneously may not be effective when applied to heterogeneous populations. In this section, we first develop useful intuition on heterogeneous early stopping. We then present CLASH, our new two-stage approach. Finally, we theoretically establish that CLASH improves stopping probability if the treatment harms any group of participants without inflating the rate of unnecessary stopping.

### 3.1 Motivation

The key problem with the homogeneous approach is that it averages over potential heterogeneous groups. One solution to this problem is to run a stopping test separately on different subsets of the population, but a question remains: which subset guarantees a high stopping probability? Assume for a moment that we have prior knowledge of each participant's group $g_i$ and their CATE $\tau(g_i)$. Then, Prop. 3.1 (proved in App. E.1) establishes that for $n$ sufficiently large, the stopping test that only considers participants from harmed groups stops with probability approaching 1.

**Proposition 3.1** (Group knowledge improves early stopping). *Let $\mu^* = \mathbb{E}[\tau(g_1)|\tau(g_1) > 0]$ denote the ATE for harmed groups and $S_n^* = \{i \leq n : \tau(g_i) > 0\}$ the participants from harmed groups. Let $\lambda_n^* = \Delta_n^*/\hat{\sigma}_n^*$ denote the value of a test statistic computed only on $S_n^*$, where $\mathbb{E}\Delta_n^* = \mu^*$, $(\Delta_n^* - \mu^*)/\hat{\sigma}_n^* \xrightarrow{d} \mathcal{N}(0, 1)$, $\sqrt{|S_n^*|}\hat{\sigma}_n^* \xrightarrow{p} \sigma^*$, and $|S_n^*|$ diverges in probability to infinity. Fix any stopping threshold $b(\alpha)$. Then, the stopping test that only considers $S_n^*$ has stopping probability converging to 1, i.e., $\mathbb{P}(\lambda_n^* > b(\alpha)) \to 1$.*

The assumptions of Prop. 3.1 hold for many popular frequentist hypothesis tests, including an OF-adjusted z-test, an OF-adjusted t-test, and an OF-adjusted stopping test to detect differences in restricted mean survival time (RMST) for time-to-event data [28]. To give a rough intuition of

---

[3]We do not refer to these as statistical power and Type I error rate to prevent confusion between the hypothesis test of the overall experiment and that of the stopping test. However, we note the equivalence.

the proof, the assumed conditions imply that the mean of $\lambda_n^*$ grows with $|S_n^*|$ while its variance remains bounded, and so $\lambda_n^*$ eventually exceeds any fixed bound. Prop. 3.1 establishes that to stop the experiment with high probability, the stopping test should be run only on the set of harmed groups $S_n^*$. We emphasize that the same stopping test applied to the whole population has no such guarantees. For example, if the overall ATE is 0, then the stopping probability of the test applied homogeneously is $\leq \alpha$. Thus, running the stopping test only on $S_n^*$ stops the trial with far higher probability.

One way to apply a stopping test only to the harmed groups is to assign a weight to each observation $w_i^* = \mathbb{1}[i \in S_n^*]$, then reweight the test statistic by this indicator. For example, for the OF-adjusted z-test in (1), using the following weighted test statistic is equivalent to running the test only on $S_n^*$,

$$\lambda_n^{*OF} = \frac{\sqrt{\sum_{1=1}^{n} w_i^*}}{\sqrt{2\sigma^2}} \left( \frac{\sum_{i=1}^{n} w_i^* y_i d_i}{\sum_{1=1}^{n} w_i^* d_i} - \frac{\sum_{i=1}^{n} w_i^* y_i (1-d_i)}{\sum_{1=1}^{n} w_i^* (1-d_i)} \right). \tag{3}$$

Prop. 3.1 implies that using $\lambda_n^{*OF}$ leads to high stopping probability. We thus call $w_i^*$ the optimal weights, as using them to reweight $\lambda_n$ guarantees stopping probability close to 1 for large enough $n$.

Of course, this insight is not of direct practical use, as it assumes knowledge of $\tau(g_i)$ and $g_i$, neither of which is observed. However, it motivates how to construct better stopping tests for heterogeneous populations. If we can infer from the data which groups are harmed, then we can use this information to accelerate early stopping. Specifically, if we can estimate the optimal weights $w_i^*$, then we can use these estimates to construct weighted tests that have high stopping probability. In contrast, stopping tests applied homogeneously will have no such guarantees. This is the insight that drives our method CLASH, which we present in the following section.

## 3.2 Method

At any interim checkpoint $n$, CLASH operates in two stages. In Stage 1, CLASH uses causal machine learning to estimate the probability that each participant $i$ is harmed by the treatment. Then in Stage 2, it uses these inferred probabilities to reweight the test statistic of any chosen stopping test.

**Stage 1: Estimate $w_i^*$**     We first estimate the treatment effect $\tau(x_i)$ for each participant $i \leq n$ using any method of heterogeneous treatment effect estimation. The specific choice of method is left to the investigator; potential options include linear models, causal forests [38], and meta-learners [17].

To maintain conditional independence between the estimated CATE and the observed outcome for participant $i$, we exclude $(x_i, y_i)$ from the training set when estimating $\tau(x_i)$. We thus use $\hat{\tau}_{n,-i}(x_i)$ to denote the estimated CATE for participant $i$, and $\hat{\sigma}_{n,-i}(x_i)$ for the estimated standard error of $\hat{\tau}_{n,-i}(x_i)$ (further discussed below). While the notation suggests leave-one-out cross validation, the estimation can instead use k-fold or progressive cross-validation [5]. We then set

$$\hat{w}_i^n = 1 - \Phi\left( \frac{\delta - \hat{\tau}_{n,-i}(x_i)}{\hat{\sigma}_{n,-i}(x_i)} \right) \tag{4}$$

where $\Phi$ is the cumulative distribution function (CDF) of the standard normal and the hyperparameter $\delta > 0$ (discussed below) is a small number. Intuitively, $\hat{w}_i^n$ conveys the probability that $i$ was harmed by the treatment; we establish that it is a good estimate of the optimal $w_i^*$ in Sec. 3.3. We also discuss the choice of $\Phi$ in Sec. 3.3.

**Stage 2: Weighted Early Stopping**     We now use $\hat{w}_i^n$ to weight the test statistic of any existing stopping test. The choice of test is left to investigators based on their experiment and domain. For example, CLASH with the OF-adjusted z-test would use the test statistic from (3), replacing $w_i^*$ with $\hat{w}_i^n$. We provide examples of weighted test statistics for other stopping tests in App. C.

We make three important practical notes. First, cross-validation (CV) in Stage 1 is employed to prevent unnecessary stopping with small samples. CV ensures that if the treatment has no effect, then the predicted CATE for participant $i$ will be independent of its observed outcome. Without this independence, outcomes that are large by random chance could be assigned high weights, especially in small samples. Second, most CATE estimation methods are designed to correct for confounding, which is not required in the experimental setting since the treatment is randomly administered. This simplifies the estimation problem (see App. D). Third, the CLASH weights (4) require not only point estimates of the CATE for each participant, but also standard errors. While bootstrapping can be used to obtain standard errors for any CATE estimation method, it is often computationally expensive. To save time and resources, investigators can use methods that compute standard errors during the process of fitting (e.g., causal forests or linear models).

Finally, we briefly discuss how to set the hyperparameter $\delta$. For our theoretical guarantees to hold (Sec. 3.3), $\delta$ must be set to a number smaller than the smallest harmful treatment effect. This choice ensures that $\hat{w}_i^n$ converges to $w_i^*$ for all values of $\tau(x_i)$. In practice, we find that it is sufficient to set $\delta$ to a number that reflects the experiment's minimum effect size of interest. It is standard for clinical trials and A/B tests to specify this effect size, especially for *pre-hoc* power calculations [34]. Thus, $\delta$ can be set using the standard experimental procedures.

## 3.3 Properties

Recall that an effective method of heterogeneous early stopping must 1) have high stopping probability if any group of participants is harmed and 2) limit the rate of unnecessary stopping. CLASH achieves both of these criteria by constructing weights $\hat{w}_i^n$ that converge in probability to the optimal weights $w_i^* \ \forall \ i$ (Thm. 3.2). Thus, for $n$ large enough, CLASH is comparable to a stopping test run only on the harmed population $S_n^*$ and hence has high stopping probability in the presence of harm (Prop. 3.1). Further, if no group is harmed, the CLASH weights converge quickly to zero. Hence, the test statistic used in Stage 2 converges to zero, as does the probability of unnecessary stopping (Thm. 3.3). Thus, CLASH has high statistical power and limits the Type I error rate relative to the null hypothesis (2).

Our first theorem (proved in App. E.2) establishes the convergence of the CLASH weights to the optimal weights. It makes mild assumptions on the quality of CATE estimation in Stage 1. The choice of CATE estimator may demand further assumptions on the data; for example, using a linear regression provides suitable estimates if $\tau(x)$ is linear in $x$. Alternative causal machine learning methods minimize these assumptions; for example, a causal forest provides sufficiently accurate estimates under weak regularity conditions (e.g., bounded $x$ [38, Thm. 4.1]).

**Theorem 3.2** (CLASH weights converge to optimal weights). *CLASH weights* (4) *satisfy the error bound*

$$|\hat{w}_i^n - w_i^*| \leq \exp\left\{ -\frac{(\tau(x_i)-\delta)^2}{2\hat{\sigma}_{n,-i}^2(x_i)} \right\} + \frac{|\hat{\tau}_{n,-i}(x_i)-\tau(x_i)|}{\hat{\sigma}_{n,-i}(x_i)} \exp\left\{ -\frac{(\tau(x_i)-\delta)^2}{2\hat{\sigma}_{n,-i}^2(x_i)} + \frac{|\tau(x_i)-\delta||\hat{\tau}_{n,-i}(x_i)-\tau(x_i)|}{\hat{\sigma}_{n,-i}^2(x_i)} \right\}.$$

*Moreover, if $\delta < \inf_{x:\tau(x)>0} \tau(x)$ and, given $x_i$, $\hat{\tau}_{n,-i}(x_i) \xrightarrow{p} \tau(x_i)$ and $\hat{\sigma}_{n,-i}(x_i) \xrightarrow{p} 0$, then $\hat{w}_i^n$ is a consistent estimator of the optimal weight: $\hat{w}_i^n - w_i^* \xrightarrow{p} 0$.*

Thm. 3.2 provides important intuition on when CLASH will perform well. For observation $i$, the rate of weight convergence depends on two factors. First, it is proportional to $\tau(x_i)$, the size of the harmful effect. The more harmful the treatment, the better CLASH will perform. Second, it is inversely proportional to $\hat{\sigma}_{n,-i}(x_i)$, the uncertainty in the estimate of $\tau(x_i)$. The more certain the estimate, the better CLASH will perform. Many factors impact $\hat{\sigma}_{n,-i}(x_i)$, including $n$ and the frequency of observing covariates like $x_i$. We expect CLASH to perform better with (1) larger datasets, (2) fewer covariates, and (3) larger $S_n^*$. We explore these relationships further in our experiments (Sec. 4-5).

We emphasize that Thm. 3.2 implies a fast rate of convergence, as the error bound decays exponentially in $1/\hat{\sigma}_{n,-i}^2(x_i)$. For example, if $\hat{\sigma}_{n,-i}^2(x_i) = o_p(1/\log n)$, then $|\hat{w}_i^n - w_i^*| = o_p(1/n)$ (see Cor. F.1). This is extremely fast: weights and other nuisance parameters are often estimated at a rate of $1/\sqrt{n}$ or slower [6]. This speed justifies our use of the Gaussian CDF in (4). While other CDFs can provide Thm. 3.2's consistency result, they may not demonstrate such provably fast convergence. Note that the additional assumption on $\hat{\sigma}_{n,-i}^2(x_i)$ is mild,[4] as it just needs to decay faster than $1/\log n$. For example, a linear regression under standard assumptions (as in [36]) has $\hat{\sigma}_{n,-i}(x_i) = \mathcal{O}_p(1/\sqrt{n})$.

Our next theorem (proved in App. E.3) establishes that CLASH limits the rate of unnecessary stopping for the OF-adjusted z-test. If no group of participants is harmed, CLASH's stopping probability converges to zero. The rate of unnecessary stopping (i.e., Type I error rate relative to (2)) is thus guaranteed to be $\leq \alpha$ for $n$ large enough. The theorem requires a mild condition on the decay of $\max_{i \leq n} \hat{\sigma}_{n,-i}^2(x_i)$ (the same discussed above). It also assumes bounded outcomes, a condition satisfied in most real experiments. Thm. F.2 provides a similar result for an SPRT stopping test.

**Theorem 3.3** (CLASH limits unnecessary stopping). *Consider a stopping test with weighted z-statistic* (3) *and weights estimated using CLASH. If $\max_{i \leq n} \hat{\sigma}_{n,-i}^2(x_i) = o_p(1/\log(n))$, $\max_{i \leq n} |\tau(x_i) - \hat{\tau}_{n,-i}(x_i)| = o_p(1)$, and $y_i$ are uniformly bounded, then the stopping probability of the test converges to zero if no participant group is harmed.*

---

[4]See [6] for a discussion of the rate of convergence of machine learning-based CATE estimators.

We note that our presented theory is asymptotic, not exact. However, many stopping tests—both frequentist and sequential—rely on asymptotic approximations and so large enough sample sizes are often already required. For example, the mSPRT stopping test with binary outcomes in [14] requires a large-sample Gaussian approximation. In our simulation experiments (Sec. 4) and real-world application (Sec. 5), we show that sample sizes typical in many clinical trials and A/B tests are sufficient for CLASH to improve stopping probability while limiting unnecessary stopping.

### 3.4 The Decision to Stop

Before proceeding to our experimental evaluation, we emphasize that CLASH is not intended to automate human decisions on early stopping. When CLASH indicates that an experiment should be stopped, the investigator can make one of several decisions. If the nature of harm is serious (e.g., mortality in a clinical trial), the investigator may decide to stop the experiment for all participants. For milder harms (e.g., high latency during application startup in an A/B test), they may decide to stop the experiment only for the harmed group and continue it for all other participants. If the identified harm is much less consequential than the potential benefit (e.g., harm of increased headaches vs. benefit of curing cancer), the investigator may decide to not stop the experiment at all. The specific decision must depend heavily on the ethical and financial aspects of the experiment, a thorough review of the interim data, guidance from the investigator's institutional review board, and other relevant factors.

If the investigator chooses to stop the experiment only on the harmed group, they face two practical challenges: identifying the harmed group and ATE estimation. To identify the harmed group, investigators can either use existing subgroup identification methods (e.g., [1, 30, 44]) on the observed outcomes or analyze the CLASH weights to find covariate combinations with weights close to 1. To estimate the whole population ATE at the end of the experiment, investigators can use inverse propensity weights; this approach allows one to correct for the selection bias induced by stopping the experiment only in one group (see example in Tab. S6). Further details, including ensuring actionable group sizes, are discussed in App. G.

## 4 Simulation Experiments

We now assess the performance of CLASH in an extensive series of simulation experiments.[5] We focus on two important types of experimental outcomes: Gaussian and time-to-event (TTE). Gaussian outcomes are the most frequently considered (e.g., [14]), as they are common in both clinical trials and A/B tests. Many stopping tests assume either that the data is Gaussian or that the average outcome is asymptotically Gaussian. Meanwhile, TTE outcomes are very common in clinical trials. We establish that CLASH improves stopping times over relevant baselines in both these scenarios. Our real-world application (Sec. 5) then applies CLASH to outcomes that are approximately negative binomial, further demonstrating that CLASH can be impactful in experiments across several domains.

**Setup: Gaussian Outcomes**    We consider a randomized experiment that evaluates the effect of $D$ on $Y$. Participants come from two groups, with $G \in \{0, 1\}$ indicating membership in a minority group. We do not observe $G$, but observe a set of binary covariates $X = [X_1, .., X_d]$, where $d$ varies between 3 and 10 and $p(X_j = 1) = 0.5 \; \forall \; j$. $X$ maps deterministically to $G$, with $G = \prod_{j=1}^{k} X_j$.[6] We vary $k$ between 1 and 3: the expected size of the minority thus varies between 12.5% and 50% (of all participants). $Y$ is normally distributed within each group, with $Y|G = 0 \sim \mathcal{N}(\theta_0 D, 1)$ and $Y|G = 1 \sim \mathcal{N}(\theta_1 D, 1)$. The distribution of $Y$ over the whole population is thus a mixture of two Gaussians. We vary $\theta_0$ between 0 and -0.1; the majority group is thus unaffected or benefited by the treatment. We vary $\theta_1$ between 0 and 1: the minority group is thus either unaffected or harmed.

The experiment runs for $N = 4000$ participants and $T = 2000$ time steps, recruiting one treated and one untreated participant at each step.[7] The experiment has three checkpoints, with 1,000, 2,000, and 3,000 participants (corresponding to 25%, 50%, and 75% of the total time duration). At each

---

[5]Replication code for the simulation experiments can be found at https://github.com/hammaadadam1/clash

[6]This is a realistic setting (e.g., $X$ is device type and new feature $D$ increases application latency $Y$ only on some devices). However, CLASH also performs well if the mapping from X to G is stochastic (see App. H.1.9).

[7]This assumption is required for some stopping tests (e.g., mSPRT, see [14]), not for CLASH. CLASH can be used with any experiment design (e.g. batch recruitment, imbalanced arms) if a suitable stopping test is used.

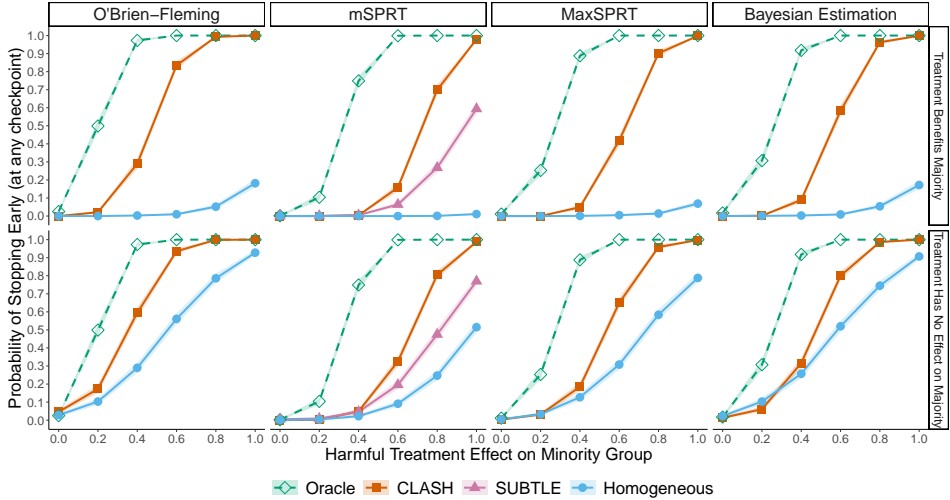

Figure 2: Performance of CLASH in a simulation experiment with Gaussian outcomes. If the minority group is harmed, CLASH (red) significantly increases the stopping probability over the homogeneous baseline (blue) and SUBTLE (pink). For large effect sizes, CLASH is as effective as an oracle that has prior knowledge of the harmed group (green). Crucially, if the treatment has no harmful effect (i.e., effect = 0), CLASH does not stop the trial more often than either baseline (i.e., CLASH limits Type I error rate). These results hold for four commonly-used stopping tests (corresponding to each column). SUBTLE builds upon the mSPRT framework specifically, so is only presented in the second column. These results hold whether the majority group's treatment is beneficial (top row) or has no effect (bottom row). We simulate 1,000 trials with 4,000 participants each and a 12.5% minority group; we plot the stopping probability (with 95% CIs) at any interim checkpoint.

checkpoint, we run CLASH and compute its stopping probability across 1,000 replications. In Stage 1, CLASH uses a causal forest with 5-fold CV and $\delta = 0.1$. Standard errors were estimated by the causal forest itself (i.e., not via the bootstrap). In Stage 2, CLASH uses one of four commonly-used stopping tests: an OF-adjusted z-test, an mSPRT [14], a MaxSPRT [16], and a Bayesian estimation-based test.

We compare CLASH's stopping probability to three alternative approaches (two baselines and one oracle). The **homogeneous baseline** applies the four aforementioned stopping tests to the collected data in aggregate. The **SUBTLE baseline** uses a recently-developed heterogeneous stopping test [43]. SUBTLE is the only existing approach that can handle unknown groups without strong parametric assumptions (e.g., linearity). SUBTLE builds on the mSPRT framework and has the same decision rule; we thus compare its performance to CLASH using mSPRT in Stage 2. The **Oracle** applies the four aforementioned stopping tests only to data from the harmed group. This approach reflects the optimal test discussed in Prop. 3.1, and represents an upper bound on performance.

**Setup: TTE Outcomes**    We consider a clinical trial that measures time to a positive event (e.g., remission) and defines treatment effects using the hazard ratio. We adapt the simulation setup from [20]; see App. H.2.1 for a full description. CLASH's Stage 1 uses a survival causal forest [8] and Stage 2 uses an OF-adjusted Cox regression. Note that existing SPRTs cannot be applied to TTE outcomes, as they cannot account for repeated observations for each participant. Crucially, SUBTLE cannot be used either; CLASH is thus the first heterogeneous method applicable to this setting.

**Results**    CLASH outperforms both the homogeneous baseline and SUBTLE in a wide range of experiments. We first discuss performance in the Gaussian setting with five covariates and a minority group that represents 12.5% of participants. Fig. 2 plots the probability that the experiment is stopped at any of the three checkpoints. If the minority group is harmed (x-axis $\theta_1 > 0$), CLASH stops the experiment significantly more often than the homogeneous and SUBTLE baselines. The magnitude of the improvement depends on the effect size: the larger the effect, the better CLASH performs. For large effects, CLASH even converges to the oracle. Crucially, CLASH does not increase the rate of unnecessary stopping: if the minority group is unharmed ($\theta_1 = 0$), CLASH stops the experiment no more frequently than either baseline (i.e., CLASH controls the Type I error rate). These results

hold for all four stopping tests; CLASH thus leads to effective stopping no matter which test is used. Moreover, its performance gains are robust to the specific choice of the hyperparameter $\delta$ (Fig. S5).

We now compare CLASH to SUBTLE, the heterogeneous baseline presented in the second column of Fig. 2. CLASH improves stopping probability over SUBTLE whether the treatment benefits the majority group or has no effect. We focus on the former case and plot the increase in stopping probability of CLASH (using mSPRT) over SUBTLE in Fig. 3. CLASH improves stopping probability in a large number of scenarios. We make three specific notes. First, when the minority group is larger, CLASH performs well for a broader range of effect sizes. Notably, SUBTLE greatly underperforms CLASH for medium effects (though it is able to detect large effects as often as CLASH). Second, increasing the number of covariates only has a small impact on CLASH, which outperforms SUBTLE despite the higher dimensionality. Third, CLASH greatly increases the probability of stopping the trial at the first and second checkpoints. CLASH thus not only stops trials more often, but also faster.

CLASH also demonstrates strong performance with TTE outcomes (App. H.2). CLASH improves the stopping probability of the OF-adjusted Cox regression by 20 percentage points if the treatment harms 25% of the population with hazard ratio of 0.7. No existing method for heterogeneous stopping (including SUBTLE) can handle TTE; this broad applicability is one of CLASH's major advantages.

App. H presents detailed simulation results that further demonstrate CLASH's efficacy. We note the discussed results are robust across a range of simulation settings, including effect size, sample size, covariate dimensionality, harmed group size, and outcome variance. Furthermore, CLASH is highly effective in situations with multiple harmed groups, noisy group membership, and unknown outcome variance. While CLASH broadly outperforms existing methods, we note two specific limitations. First, CLASH may struggle to detect harmful effects in very small samples (e.g, N=200, see Fig. S10). Note that this is a challenging setting for any method, since we only observe 10-20 data points from the harmed group at any interim checkpoint. Second, CLASH may struggle with a very large number of covariates (e.g., 500 covariates, see Fig. S11). In such settings, we recommend performing feature selection before running CLASH.

## 5 Real-world Application

In this section, we apply CLASH to real data from a digital experiment. We consider an A/B test run by a technology company to evaluate the effect of a software update on user experience. The dataset was collected in 2022 and consists of 500,000 participants, roughly half of whom received the update.

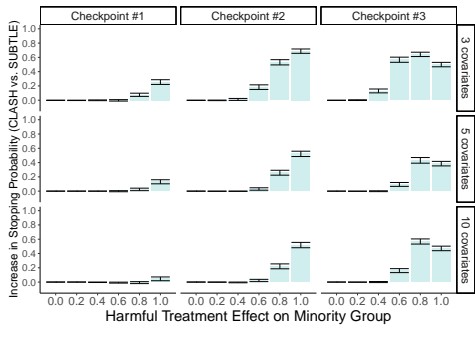
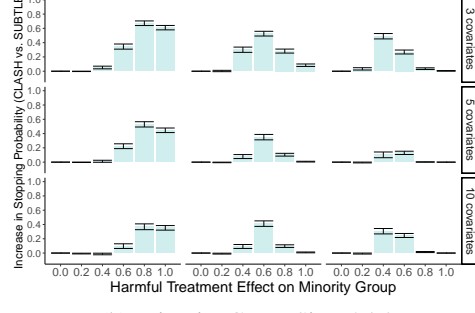

(a) Minority Group Size: 12.5%
(b) Minority Group Size: 25%

Figure 3: CLASH's performance improvement over the SUBTLE baseline with Gaussian outcomes. CLASH significantly increases stopping probability over SUBTLE across a range of effect sizes (x-axis) and covariate dimensions (rows). CLASH not only stops more often but also faster, with higher stopping probability at earlier checkpoints (columns). CLASH does not increase stopping probability if the treatment is not harmful (i.e., effect = 0). We plot the difference in stopping probability between CLASH and SUBTLE (with 95% CIs), where the minority group forms (a) 12.5% or (b) 25% of all participants. For the larger group size, CLASH's improvement is most notable for medium effects, which SUBTLE struggles to detect. All performance increases are robust to an increase in the number of covariates. See App. H.1 for a similar comparison of CLASH to the homogeneous baseline.

We evaluate this treatment's effect on a proprietary metric that measures negative user experience. An increase in the outcome—which is discrete and right-skewed—indicates a worse user experience. The experiment also recorded which one of eight geographic regions the user's device was located in. We first use the full dataset to assess whether the treatment had a heterogeneous impact by region. Separate negative binomial (NB) regressions indicate that the treatment led to a large statistically significant increase in the metric in Region 1, a small significant increase in Regions 2, 3, and 4, no significant effect in Regions 5 and 6, and a significant decrease in Regions 7 and 8 (Tab. S3).

We evaluate stopping tests on this dataset, conducting checkpoints every 20,000 participants. We run CLASH with a causal forest and $\delta = 1$ in Stage 1 and an OF-adjusted NB regression in Stage 2. We find that CLASH decreases stopping time over the homogeneous baseline, stopping the experiment after 40,000 as opposed to 60,000 participants. SUBTLE stops after 80,000 participants; this under-performance may result from the data's strong non-normality. An oracle that only considers Region 1 also stops after 40,000 observations (i.e., same as CLASH). We note that CLASH's optimal performance is not a fluke: across 1,000 random shuffles of the dataset, CLASH stops the experiment at the same interim checkpoint as the Oracle in 62.6% of shuffles (details in Tab. S5).

We emphasize that by stopping the experiment early, the company can minimize customer frustration, preventing churn and financial impact. Domain expertise can guide whether to stop the experiment altogether, or just in the harmed regions. The harmed regions can be identified at stopping time using separate regressions (Tab. S4), CLASH weights (Fig. S19), or policy learning [15, 1]. Note that stopping the experiment just in one region would affect statistical inference at the end of the experiment, as the treatment would no longer be randomly assigned across regions. As described in Sec. 3.4, investigators can use inverse probability weighting to adjust for this selection (Tab. S6).

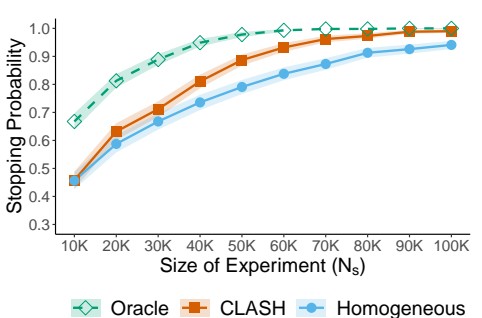

Figure 4: CLASH performance by sample size on real data. We plot stopping probability across 1,000 experiments with 95% CIs.

Finally, we study the impact of sample size on CLASH's performance. We vary sample size $N_s$ between 10,000 and 100,000, and generate 1,000 experiments for each $N_s$ by sampling from our dataset. We only sample from Region 1 and Regions 5-8; this gives us one harmed group (Region 1) that comprises 29% of the total population (see App. I for results with Regions 2-4). We conduct one checkpoint at the halfway stage of each simulated experiment. We find that CLASH significantly improves stopping probability over the homogeneous baseline for all $N_s \geq 40,000$, and converges to near-oracle performance around $N_s = 90,000$ (Fig. 4). These sizes represent 8% and 18% of the total sample of the A/B test. This result is encouraging, as it indicates that datasets that are a fraction of the size of typical online A/B tests are large enough for our asymptotic theory (Sec. 3.3) to hold.

## 6 Discussion and Limitations

We propose a new method CLASH for the early stopping of randomized experiments on heterogeneous populations. CLASH stops experiments more often than existing approaches if the treatment harms only a subset of participants, and does not stop experiments if the treatment is not harmful. Prior work is either limited by restrictive assumptions (e.g., linearity) or incompatible with common data distributions (e.g., time-to-event). In contrast, CLASH is easy to use, adapts any existing stopping test, and is broadly applicable to clinical trials and A/B tests. Our work has a few important limitations. First, CLASH works better with a relatively small number of covariates. While this is not a problem for most experiments—clinical trials and A/B tests typically collect only a few covariates—a different pre-processing approach may be required in high-dimensional settings (e.g., genetic data with over 500 covariates). Second, CLASH may not detect harm in experiments with a very small number of participants. For example, sample sizes typical in Phase 1 clinical trials (i.e., less than 100 participants) may be too small for CLASH to be more effective than existing approaches. Finally, this paper only considers stopping for harm. In practice, experiments are often stopped for early signs of benefit and futility. Expanding CLASH to these decisions is an important direction for future work.

## Acknowledgments and Disclosure of Funding

We thank Steven Goodman, Sanjay Basu, Dean Eckles, Nikos Trichakis, and Sonia Jaffe for their helpful comments. This work was graciously supported through the Bowers CIS Strategic Partnership Program with LinkedIn.

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

## A    Early Stopping Methods

Table S1: Comparison of stopping tests. We categorize commonly-used stopping tests into three categories—frequentist, sequential, and Bayesian—and describe their key features. Stopping type describes whether the stopping test is used to stop for benefit (B), harm (H), or futility (F). Stopping criteria refers to the statistic that is used to make the stopping decision. Monitoring frequency describes how often the method allows peeking at the data; for example, frequentist methods only allow for a small number of interim checkpoints, while sequential methods allow for continuously monitoring outcomes as they are observed. Finally, the test type describes what kinds of alternate hypotheses the method allows for testing; the SPRT, for example, only allows the investigator to test simple vs. simple hypotheses.

| Stopping Test | Stopping Type | Stopping Criteria | Monitoring Frequency | Test Types |
|---|---|---|---|---|
| **Frequentist / group sequential** | | | | |
| Pocock [23] | B/H | p-value | Limited checks, pre-planned | All |
| O'Brien-Fleming [22] | B/H | p-value | Limited checks, pre-planned | All |
| Alpha-spending [10] | B/H | p-value | Limited checks, flexible | All |
| Conditional power [18] | F | Conditional power | Limited checks, flexible | All |
| **Sequential / quasi-Bayesian** | | | | |
| SPRT [39] | B/H/F | Likelihood ratio | Continuous | Simple |
| 2-SPRT [19] | B/H/F | Likelihood ratio | Continuous | One-sided |
| Max-SPRT [16] | B/H | Likelihood ratio | Continuous | Two-sided |
| Mixture-SPRT [26, 14] | B/H | Likelihood ratio / always-valid p-value | Continuous | Two-sided |
| Martingale-based [3, 13] | B/H | Always-valid p-value | Continuous | All |
| **Bayesian** | | | | |
| Parameter estimation [33] | B/H | Posterior expected loss | Continuous | All |
| Hypothesis testing [11] | B/H | Posterior odds | Continuous | All |
| Predictive probability [29] | F | Posterior predictive probability | Continuous | All |

## B    Adapting the SPRT for Early Stopping

Here, we briefly describe how to adapt SPRT-based tests for early stopping. This largely follows from Section 4.3 in [14]. Recall that for every participant $i$, we observe their treatment assignment $d_i$, outcome $y_i$, and covariates $x_i$. We now slightly update this notation. Assume that at time $t$, we observe two participants, one treated, one untreated. Let $y_t^{(1)}$ and $y_t^{(0)}$ denote the outcomes for the treated and untreated participants respectively, and $x_t$ denote their joint covariates. We define the difference in outcomes observed at time $t$, $z_t = y_t^{(1)} - y_t^{(0)}$.

We can now use Wald's SPRT [39], mSPRT [14], and other sequential tests in this setup. For example, assume that $z_t$ follows a known distribution with probability density function $p$ and mean parameter $\mu$. Then, to test between the null hypothesis of no effect $H_0 : \mu = 0$ against an alternate hypothesis of a specific harmful effect $H_1 : \mu = \beta$, Wald's SPRT would use

$$\lambda_{2n}^{SPRT} = \sum_{t=1}^{n} \log \frac{p(z_t|\mu = \beta)}{p(z_t|\mu = 0)}$$

$$b^{SPRT}(\alpha) = -\log \alpha$$

Note that we denote the statistic as $\lambda_{2n}^{SPRT}$, not $\lambda_n^{SPRT}$, as it uses $2n$ observations ($n$ treated, $n$ untreated). For the special case where $p$ is the Gaussian probability density function, the test statistic reduces to

$$\lambda_{2n}^{SPRT} = \beta \sum_{t=1}^{n} z_t - \frac{n}{2}\beta^2.$$

## C Weighted Test Statistics

We present examples of weighted test statistics for some commonly-used stopping tests in Tab. S2. Note that weighted inference is already implemented in standard data analysis packages for a wide range of estimators. For example, for a group-sequential test (e.g., O'Brien-Fleming) that considers p-values from a generalized linear model, standard packages in R [24] and Python [31] allow for user-defined regression weights.

Table S2: Weighted test statistics for commonly used stopping tests.

| Stopping Test | Test Statistic | Weighted Test Statistic |
|---|---|---|
| Group-sequential z-test | $\frac{\sqrt{n}}{\sqrt{2\sigma^2}}\left(\frac{\sum_{i=1}^n y_i d_i}{\sum_{1=1}^n d_i} - \frac{\sum_{i=1}^n y_i(1-d_i)}{\sum_{1=1}^n (1-d_i)}\right)$ | $\frac{\sqrt{\sum_{1=1}^n w_i}}{\sqrt{2\sigma^2}}\left(\frac{\sum_{i=1}^n w_i y_i d_i}{\sum_{1=1}^n w_i d_i} - \frac{\sum_{i=1}^n w_i y_i(1-d_i)}{\sum_{1=1}^n w_i(1-d_i)}\right)$ |
| Generic SPRT | $\sum_{t=1}^n \log \frac{p(z_t\mid\mu=\beta)}{p(z_t\mid\mu=0)}$ | $\sum_{t=1}^n w_t \log \frac{p(z_t\mid\mu=\beta)}{p(z_t\mid\mu=0)}$ |
| Gaussian SPRT | $\beta\sum_{t=1}^n z_t - \frac{n}{2}\beta^2$ | $\beta\sum_{t=1}^n w_t z_t - \frac{\sum_{t=1}^n w_t}{2}\beta^2$ |
| Gaussian mSPRT [14] | $\sqrt{\frac{2\sigma^2}{2\sigma^2+\tau^2 n}}\exp\left\{\frac{\tau^2(\sum_{t=1}^n z_t - \theta_0 n)^2}{4\sigma^2(2\sigma^2+\tau^2 n)}\right\}$ | $\sqrt{\frac{2\sigma^2}{2\sigma^2+\tau^2\sum_{t=1}^n w_t}}\exp\left\{\frac{\tau^2(\sum_{i=1}^n w_t z_t - \theta_0\sum_{t=1}^n w_t)^2}{4\sigma^2(2\sigma^2+\tau^2\sum_{t=1}^n w_t)}\right\}$ |
| Gaussian maxSPRT (based on [16]) | $\frac{\sum_{t=1} z_t}{n}\sum_{t=1}^n z_t - \frac{n}{2}\left(\frac{\sum_{t=1} z_t}{n}\right)^2$ | $\frac{\sum_{t=1} w_t z_t}{\sum_{t=1} w_t}\sum_{t=1}^n w_t z_t - \frac{\sum_{t=1} w_t}{2}\left(\frac{\sum_{t=1} w_t z_t}{\sum_{t=1} w_t}\right)^2$ |

## D CATE Estimation with No Confounding

Most CATE estimation methods are designed to control for confounders, that is, variables that affect both the treatment status $D$ and observed outcome $Y$. Our work, however, considers experiments in which $D$ is randomly administered; there is no confounding, as $D$ is independent of all variables except $Y$. The CATE estimation method thus only needs to infer how the treatment effect varies with the covariates $X$ and does not need to correct for the relationship between $X$ and $D$. This simplifies the application of several CATE estimation methods; we discuss a few specific examples below. In short, many methods use estimates of the treatment propensity $\eta(x) = \mathbb{P}(D = 1 \mid X = x)$ to correct for confounding. In our setting, we can simply set $\eta(x) = 0.5 \ \forall \ x$.

- **Causal forests**    While the original causal forest algorithm [38] did not estimate a propensity score, later improvements [2, 21] use estimates of $\eta(x)$ to "orthogonalize" the estimator. In practice, this orthogonalization is essential in obtaining accurate CATE estimates from observational data [2]. However, it is not required in our experimental setting; these improved algorithms [2, 21] can be used by setting $\eta(x)$ to its true value (i.e., 0.5).

- **Meta-learners**    T-learners and S-learners (as described in [17]) do not use estimates of the propensity score and thus do not require any modifications. The X-learner [17] uses an estimate of $\eta(x)$ to weight the predictions of models trained separately on the treatment and control groups [see 17, equation 9]. These weights can be replaced by 0.5 in our setting.

- **Linear Models**    Linear CATE models often use estimates of $1/\eta(x)$ as regression weights. This approach ensures that the estimation is "doubly robust": as long as either the model of treatment assignment or outcome heterogeneity is correctly specified, the resulting regression coefficients are unbiased [27]. In our setting, such reweighting is unnecessary, as $1/\eta(x)$ is known and the same for all observations.

Not needing to estimate the propensity score has two key benefits. First, it reduces the computational complexity of the problem. Repeated fitting techniques like cross-fitting [6] are often used to reduce the bias of simultaneous propensity score and treatment effect estimation, but these procedures increase the computational cost of CATE estimation, especially for large datasets. Second, it protects against inaccurate estimates of the treatment propensity. Poor estimates of $\eta(x)$ can lead to highly biased treatment effect estimates [12]. By explicitly specifying $\eta(x) = 0.5$, we avoid spuriously inferring any relationships between $X$ and $D$ that would create such bias.

# E  Proofs

We now present proofs for all theoretical results described in the main text. Note that we repeatedly use the following property. For any events $A$ and $B$,

$$\mathbb{P}(A, B) \leq \min\{\mathbb{P}(A), \mathbb{P}(B)\}. \tag{5}$$

## E.1  Proof of Prop. 3.1: Group knowledge improves early stopping

We argue that the stopping probability of the test on $S_n^*$ converges to 1. Recall that

$$\lambda_n^* = \frac{\Delta_n^*}{\hat{\sigma}_n^*}.$$

By assumption, $\lambda_n^* - \mu^*/\hat{\sigma}_n^* \xrightarrow{d} \mathcal{N}(0,1)$ and $\sqrt{|S_n^*|}\hat{\sigma}_n^*/\sigma^* \xrightarrow{p} 1$. Define $\rho_n^* = \sqrt{|S_n^*|}\Delta_n^*/\sigma^*$. Then, by Slutsky's theorem,

$$\rho_n^* - \mu^*\sqrt{|S_n^*|}/\sigma^* \xrightarrow{d} \mathcal{N}(0,1).$$

Now, consider the probability that the test does not stop. For notational convenience, fix some $\alpha > 0$ and denote $b(\alpha)$ simply as $b$. Then,

$$
\begin{aligned}
\mathbb{P}(\lambda_n^* \leq b) &= \mathbb{P}(\rho_n^* \leq b\rho_n^*/\lambda_n^*) \\
&= \mathbb{P}(\rho_n^* \leq b\rho_n^*/\lambda_n^*,\, \rho_n^*/\lambda_n^* \geq 0.5) + \mathbb{P}(\rho_n^* \leq b\rho_n^*/\lambda_n^*,\, \rho_n^*/\lambda_n^* < 0.5) \\
&\leq \mathbb{P}(\rho_n^* \leq b/2) + \mathbb{P}(\rho_n^*/\lambda_n^* < 0.5)
\end{aligned}
\tag{6}
$$

Now, we know that,

$$
\begin{aligned}
\frac{\rho_n^*}{\lambda_n^*} &= \frac{\sqrt{|S_n^*|}\Delta_n^*/\sigma^*}{\Delta_n^*/\hat{\sigma}_n^*} \\
&= \frac{\sqrt{|S_n^*|}\hat{\sigma}_n^*}{\sigma^*} \\
&\xrightarrow{p} 1.
\end{aligned}
$$

Hence, $\mathbb{P}(\rho_n^*/\lambda_n^* < 0.5) \to 0$. This bounds the second term in (6). We now focus on the first term. Fix any $\epsilon > 0$. Since $|S_n^*| \to \infty$, there must exist some $n_\epsilon$ such that $b/2 - \mu^*\sqrt{|S_n^*|}/\sigma^* < \Phi^{-1}(\epsilon)$ for all $n > n_\epsilon$. Assume $n > n_\epsilon$. Then,

$$
\begin{aligned}
\mathbb{P}(\rho_n^* \leq b/2) &= \mathbb{P}(\rho_n^* - \mu^*\sqrt{|S_n^*|}/\sigma^* \leq b/2 - \mu^*\sqrt{|S_n^*|}/\sigma^*) \\
&\leq \mathbb{P}(\rho_n^* - \mu^*\sqrt{|S_n^*|}/\sigma^* \leq \Phi^{-1}(\epsilon)) \\
&\to \Phi(\Phi^{-1}(\epsilon)) = \epsilon
\end{aligned}
$$

since $\rho_n^* - \mu^*\sqrt{|S_n^*|}/\sigma^* \xrightarrow{d} \mathcal{N}(0,1)$. Since this holds for any value of $\epsilon$, we have that $\mathbb{P}(\rho_n^* \leq b/2) \to 0$. Hence, from (6), we have that $\mathbb{P}(\lambda_n^* \leq b) \to 0$, and thus

$$\mathbb{P}(\lambda_n^* > b) \to 1$$

Hence, the stopping probability of the test on the harmed population converges to 1.

## E.2  Proof of Thm. 3.2: CLASH weights converge to optimal weights

Our entire proof will be carried out conditional on the value $x_i$. Recall that we define our weights as

$$\hat{w}_i^n = 1 - \Phi\Big(\frac{\delta - \hat{\tau}_{n,-i}(x_i)}{\hat{\sigma}_{n,-i}(x_i)}\Big)$$

Further recall that the functions $\hat{\tau}_{n,-i}$ and $\hat{\sigma}_{n,-i}$ are, by construction, independent of $x_i$.

**Error Bound** We first establish a bound on the difference between our estimated weights $\hat{w}_i^n$ and the optimal weights $w_i^*$. Consider two cases.

**Case 1:** $\tau(x_i) \leq 0$. In this case, $w_i^* = 0$ by definition, and so we just need to prove a bound on the magnitude of $\hat{w}_i^n$. Using Taylor's theorem with Lagrange remainder, we know that $\exists h_n \in [0,1]$ such that

$$
\begin{aligned}
\hat{w}_i^n &= 1 - \Phi\Big(\frac{\delta - \hat{\tau}_{n,-i}(x_i)}{\hat{\sigma}_{n,-i}(x_i)}\Big) \\
&= 1 - \Phi\Big(\frac{\delta - \tau(x_i)}{\hat{\sigma}_{n,-i}(x_i)}\Big) - \frac{\tau(x_i) - \hat{\tau}_{n,-i}(x_i)}{\hat{\sigma}_{n,-i}(x_i)}\phi\Big(\frac{\delta - \tau(x_i)}{\hat{\sigma}_{n,-i}(x_i)} + h_n\frac{\tau(x_i) - \hat{\tau}_{n,-i}(x_i)}{\hat{\sigma}_{n,-i}(x_i)}\Big) \\
&\leq 1 - \Phi\Big(\frac{\delta - \tau(x_i)}{\hat{\sigma}_{n,-i}(x_i)}\Big) + \frac{|\tau(x_i) - \hat{\tau}_{n,-i}(x_i)|}{\hat{\sigma}_{n,-i}(x_i)}\phi\Big(\frac{\delta - \tau(x_i)}{\hat{\sigma}_{n,-i}(x_i)} + h_n\frac{\tau(x_i) - \hat{\tau}_{n,-i}(x_i)}{\hat{\sigma}_{n,-i}(x_i)}\Big)
\end{aligned}
$$

where $\phi$ is the standard Gaussian probability density function. Since $\delta - \tau(x_i) > 0$, we can use the Chernoff inequality to bound the first term,

$$
1 - \Phi\Big(\frac{\delta - \tau(x_i)}{\hat{\sigma}_{n,-i}(x_i)}\Big) \leq \exp\Big(-\frac{(\delta - \tau(x_i))^2}{2\hat{\sigma}_{n,-i}^2(x_i)}\Big).
$$

We now focus on the second term, which contains

$$
\begin{aligned}
\phi\Big(\frac{\delta - \tau(x_i)}{\hat{\sigma}_{n,-i}(x_i)} &+ h_n\frac{\tau(x_i) - \hat{\tau}_{n,-i}(x_i)}{\hat{\sigma}_{n,-i}(x_i)}\Big) \\
&= \frac{1}{\sqrt{2\pi}}\exp\Big\{-\frac{1}{2}\Big(\frac{\delta - \tau(x_i)}{\hat{\sigma}_{n,-i}(x_i)} + h_n\frac{\tau(x_i) - \hat{\tau}_{n,-i}(x_i)}{\hat{\sigma}_{n,-i}(x_i)}\Big)^2\Big\} \\
&= \frac{1}{\sqrt{2\pi}}\exp\Big\{-\frac{1}{2}\Big[\Big(\frac{\delta - \tau(x_i)}{\hat{\sigma}_{n,-i}(x_i)}\Big)^2 + h_n^2\Big(\frac{\tau(x_i) - \hat{\tau}_{n,-i}(x_i)}{\hat{\sigma}_{n,-i}(x_i)}\Big)^2 - \\
&\qquad\qquad 2h_n\frac{(\delta - \tau(x_i))(\hat{\tau}_{n,-i}(x_i) - \tau(x_i))}{\hat{\sigma}_{n,-i}^2(x_i)}\Big]\Big\} \\
&\leq \exp\Big\{-\frac{1}{2}\Big[\Big(\frac{\delta - \tau(x_i)}{\hat{\sigma}_{n,-i}(x_i)}\Big)^2 - 2h_n\frac{(\delta - \tau(x_i))|\hat{\tau}_{n,-i}(x_i) - \tau(x_i)|}{\hat{\sigma}_{n,-i}^2(x_i)}\Big]\Big\} \\
&\leq \exp\Big\{-\frac{1}{2}\Big[\Big(\frac{\delta - \tau(x_i)}{\hat{\sigma}_{n,-i}(x_i)}\Big)^2 - 2\frac{(\delta - \tau(x_i))|\hat{\tau}_{n,-i}(x_i) - \tau(x_i)|}{\hat{\sigma}_{n,-i}^2(x_i)}\Big]\Big\},
\end{aligned}
$$

since $h_n \in [0,1]$. Thus, we have that

$$
\hat{w}_i^n \leq \exp\Big\{-\frac{(\delta - \tau(x_i))^2}{2\hat{\sigma}_{n,-i}^2(x_i)}\Big\} + \frac{|\tau(x_i) - \hat{\tau}_{n,-i}(x_i)|}{\hat{\sigma}_{n,-i}(x_i)}\exp\Big\{-\frac{(\delta - \tau(x_i))^2}{2\hat{\sigma}_{n,-i}^2(x_i)} + \frac{(\delta - \tau(x_i))|\hat{\tau}_{n,-i}(x_i) - \tau(x_i)|}{\hat{\sigma}_{n,-i}^2(x_i)}\Big\}.
$$

**Case 2:** $\tau(x_i) > 0$. In this case, $w_i^* = 1$. Thus,

$$
\begin{aligned}
|\hat{w}_i^n - w_i^*| &= 1 - \hat{w}_i^n \\
&= \Phi\Big(\frac{\delta - \hat{\tau}_{n,-i}(x_i)}{\hat{\sigma}_i(x_i)}\Big) \\
&= \Phi\Big(\frac{\delta - \tau(x_i)}{\hat{\sigma}_{n,-i}(x_i)}\Big) + \frac{|\tau(x_i) - \hat{\tau}_{n,-i}(x_i)|}{\hat{\sigma}_{n,-i}(x_i)}\phi\Big(\frac{\delta - \tau(x_i)}{\hat{\sigma}_{n,-i}(x_i)} + h_n\frac{\tau(x_i) - \hat{\tau}_{n,-i}(x_i)}{\hat{\sigma}_{n,-i}(x_i)}\Big) \\
&= 1 - \Phi\Big(\frac{\tau(x_i) - \delta}{\hat{\sigma}_{n,-i}(x_i)}\Big) + \frac{|\tau(x_i) - \hat{\tau}_{n,-i}(x_i)|}{\hat{\sigma}_{n,-i}(x_i)}\phi\Big(\frac{\delta - \tau(x_i)}{\hat{\sigma}_{n,-i}(x_i)} + h_n\frac{\tau(x_i) - \hat{\tau}_{n,-i}(x_i)}{\hat{\sigma}_{n,-i}(x_i)}\Big) \\
&\leq \exp\Big\{-\frac{(\delta - \tau(x_i))^2}{2\hat{\sigma}_{n,-i}^2(x_i)}\Big\} + \\
&\qquad \frac{|\tau(x_i) - \hat{\tau}_{n,-i}(x_i)|}{\hat{\sigma}_{n,-i}(x_i)}\exp\Big\{-\frac{(\delta - \tau(x_i))^2}{2\hat{\sigma}_{n,-i}^2(x_i)} + \frac{(\tau(x_i) - \delta)|\hat{\tau}_{n,-i}(x_i) - \tau(x_i)|}{\hat{\sigma}_{n,-i}^2(x_i)}\Big\},
\end{aligned}
$$

using the same Taylor expansion and Chernoff bound from above.

In summary, we have established that

$$|\hat{w}_i^n - w_i^*| \leq \exp\left\{-\frac{(\delta-\tau(x_i))^2}{2\hat{\sigma}_{n,-i}^2(x_i)}\right\} + \frac{|\tau(x_i)-\hat{\tau}_{n,-i}(x_i)|}{\hat{\sigma}_{n,-i}(x_i)}\exp\left\{-\frac{(\delta-\tau(x_i))^2}{2\hat{\sigma}_{n,-i}^2(x_i)} + \frac{|\tau(x_i)-\delta||\hat{\tau}_{n,-i}(x_i)-\tau(x_i)|}{\hat{\sigma}_{n,-i}^2(x_i)})\right\}.$$

**Consistency of $\hat{w}_i^n$**  We now establish the consistency of $\hat{w}_i^n$ using the derived bound. We assume that $\delta < \inf_{x:\tau(x)>0} \tau(x)$ and, given $x_i$, $\hat{\tau}_{n,-i}(x_i) \xrightarrow{p} \tau(x_i)$ and $\hat{\sigma}_{n,-i}(x_i) \xrightarrow{p} 0$.

Define $a_i = |\tau(x_i) - \delta|/\sqrt{2}$ and $Z_{i,n} = \frac{\tau(x_i)-\hat{\tau}_{n,-i}(x_i)}{\hat{\sigma}_{n,-i}(x_i)}$. Note that $a_i$ is strictly positive and a constant (given $x_i$). From the error bound, we have that

$$|\hat{w}_i^n - w_i^*| \leq \exp(-a_i^2/\hat{\sigma}_{n,-i}(x_i)^2) + |Z_{i,n}|\exp[-a_i^2/\hat{\sigma}_{n,-i}(x_i)^2 + \sqrt{2}a_i|Z_{i,n}|/\hat{\sigma}_{n,-i}(x_i)].$$

Now, we fix any $\epsilon > 0$. Applying the law of total probability and equation (5), we find that

$$\mathbb{P}(|\hat{w}_i^n - w_i^*| > \epsilon)$$
$$\leq \mathbb{P}(\exp(-a_i^2/\hat{\sigma}_{n,-i}(x_i)^2) + |Z_{i,n}|\exp[-a_i^2/\hat{\sigma}_{n,-i}(x_i)^2 + \sqrt{2}a_i|Z_{i,n}|/\hat{\sigma}_{n,-i}(x_i)] > \epsilon)$$
$$= \mathbb{P}(\exp(-a_i^2/\hat{\sigma}_{n,-i}(x_i)^2) + |Z_{i,n}|\exp[-a_i^2/\hat{\sigma}_{n,-i}(x_i)^2 + \sqrt{2}a_i|Z_{i,n}|/\hat{\sigma}_{n,-i}(x_i)] > \epsilon,$$
$$\quad |Z_{i,n}\hat{\sigma}_{n,-i}(x_i)| > a_i/(2\sqrt{2})) +$$
$$\quad \mathbb{P}(\exp(-a_i^2/\hat{\sigma}_{n,-i}(x_i)^2) + |Z_{i,n}|\exp[-a_i^2/\hat{\sigma}_{n,-i}(x_i)^2 + \sqrt{2}a_i|Z_{i,n}|/\hat{\sigma}_{n,-i}(x_i)] > \epsilon,$$
$$\quad |Z_{i,n}\hat{\sigma}_{n,-i}(x_i)| \leq a_i/(2\sqrt{2}))$$
$$\leq \mathbb{P}(|Z_{i,n}\hat{\sigma}_{n,-i}(x_i)| > a_i/(2\sqrt{2})) +$$
$$\quad \mathbb{P}(\exp(-a_i^2/\hat{\sigma}_{n,-i}(x_i)^2) + |Z_{i,n}|\exp[-a_i^2/\hat{\sigma}_{n,-i}(x_i)^2 + \sqrt{2}a_i|Z_{i,n}|/\hat{\sigma}_{n,-i}(x_i)] > \epsilon,$$
$$\quad |Z_{i,n}\hat{\sigma}_{n,-i}(x_i)| \leq a_i/(2\sqrt{2}))$$
$$\leq \mathbb{P}(|Z_{i,n}\hat{\sigma}_{n,-i}(x_i)| > a_i/(2\sqrt{2})) +$$
$$\quad \mathbb{P}\left(\left(1 + \frac{a_i}{2\sqrt{2}\hat{\sigma}_{n,-i}(x_i)}\right)\exp\left(\frac{-a_i^2}{2\hat{\sigma}_{n,-i}(x_i)^2}\right) > \epsilon, \; |Z_{i,n}\hat{\sigma}_{n,-i}(x_i)| \leq a_i/(2\sqrt{2})\right)$$
$$\leq \mathbb{P}(|Z_{i,n}\hat{\sigma}_{n,-i}(x_i)| > a_i/(2\sqrt{2})) + \mathbb{P}\left(\left(1 + \frac{a_i}{2\sqrt{2}\hat{\sigma}_{n,-i}(x_i)}\right)\exp\left(\frac{-a_i^2}{2\hat{\sigma}_{n,-i}(x_i)^2}\right) > \epsilon\right) \quad (7)$$

where the second-to-last inequality follows by substituting $|Z_{i,n}\hat{\sigma}_{n,-i}(x_i)| \leq a_i/(2\sqrt{2})$ into the bound and algebraically simplifying, and the last inequality follows from (5). Now, the first term on the right converges to 0 since $Z_{i,n}\hat{\sigma}_{n,-i}(x_i) = \tau(x_i) - \hat{\tau}_{n,-i}(x_i) \xrightarrow{p} 0$. We examine the second term. Define $\xi_{i,n} = a_i/(\sqrt{2}\hat{\sigma}_{n,-i}(x_i))$. Since a convex function is no smaller than its tangent line, we have $(1 + \xi_{i,n}/2) \leq \exp(\xi_{i,n}/2)$. Then,

$$\mathbb{P}\left(\left(1 + \frac{a_i}{2\sqrt{2}\hat{\sigma}_{n,-i}(x_i)}\right)\exp\left(\frac{-a_i^2}{2\hat{\sigma}_{n,-i}(x_i)^2}\right) > \epsilon\right) \leq \mathbb{P}\left((1 + \xi_{i,n}/2)\exp(-\xi_{i,n}^2) > \epsilon\right)$$
$$\leq \mathbb{P}(\exp(-\xi_{i,n}^2 + \xi_{i,n}/2) > \epsilon)$$

which converges to 0 as $\hat{\sigma}_{n,-i}(x_i) \xrightarrow{p} 0$ (and thus $\xi_{i,n}$ diverges in probability to $\infty$). Thus, we have shown that

$$\mathbb{P}(|\hat{w}_i^n - w_i^*| > \epsilon) \to 0,$$

which establishes the desired consistency.

### E.3   Proof of Thm. 3.3: CLASH limits unnecessary stopping

Suppose that no participant group is harmed so that $w_i^* = 0$ for all $i$. Let $a = \delta/\sqrt{2}$, and $a_i = |\tau(x_i) - \delta|/\sqrt{2}$, and note that $a \leq a_i$ for each $i$ as each $\tau(x_i) \leq 0$. Thus, if $|Z_{i,n}\hat{\sigma}_{n,-i}(x_i)| \leq a/(2\sqrt{2})$, then

$|Z_{i,n}\hat{\sigma}_{n,-i}(x_i)| \le a_i/(2\sqrt{2})$ and hence, as in the proof of Thm. 3.2,

$$\hat{w}_i^n \le \left(1 + \frac{a_i}{2\sqrt{2}\hat{\sigma}_{n,-i}(x_i)}\right)\exp\left(\frac{-a_i^2}{2\hat{\sigma}_{n,-i}(x_i)^2}\right).$$

Fix any $\epsilon > 0$, let $\xi_{i,n} = a_i/(\sqrt{2}\hat{\sigma}_{n,-i}(x_i))$, and define $f(x) = (1 + x/2)\exp(-x^2)$. As we did for equation (7) in the proof of Thm. 3.2, we apply the law of total probability and equation (5) to write,

$$
\begin{aligned}
\mathbb{P}(\textstyle\sum_i \hat{w}_i^n > \epsilon) \le\; & \mathbb{P}(\max_{i \le n} |Z_{i,n}\hat{\sigma}_{n,-i}(x_i)| > a/(2\sqrt{2})) + \\
& \mathbb{P}(\textstyle\sum_i \hat{w}_i^n > \epsilon \,,\, \max_{i \le n} |Z_{i,n}\hat{\sigma}_{n,-i}(x_i)| \le a/(2\sqrt{2})) \\
\le\; & \mathbb{P}(\max_{i \le n} |Z_{i,n}\hat{\sigma}_{n,-i}(x_i)| > a/(2\sqrt{2})) + \\
& \mathbb{P}(\textstyle\sum_i (1 + a_i/(2\sqrt{2}\hat{\sigma}_{n,-i}(x_i)))\exp(-a_i^2/(2\hat{\sigma}_{n,-i}(x_i)^2)) > \epsilon) \\
=\; & \mathbb{P}(\max_{i \le n} |Z_{i,n}\hat{\sigma}_{n,-i}(x_i)| > a/(2\sqrt{2})) + \mathbb{P}(\textstyle\sum_i f(\xi_{i,n}) > \epsilon).
\end{aligned}
$$

Since $\max_{i \le n} |Z_{i,n}\hat{\sigma}_{n,-i}(x_i)| = o_p(1)$ by assumption, we have $\mathbb{P}(\max_{i \le n} |Z_{i,n}\hat{\sigma}_{n,-i}(x_i)| > a/(2\sqrt{2})) \to 0$. Meanwhile, since a convex function is no smaller than its tangent line, we have $(1 + x/2) \le \exp(x/2)$ and hence

$$f(x) \le \exp(-x^2 + x/2) \le \exp(-x^2 + x^2/2 + 1/8) = \exp(-x^2/2 + 1/8)$$

where we used the arithmetic-geometric mean inequality in the final inequality. Therefore, since $a \le \min_i a_i$,

$$
\begin{aligned}
\mathbb{P}(\textstyle\sum_i f(\xi_{i,n}) > \epsilon) &\le \mathbb{P}(\textstyle\sum_i \exp(-\xi_{i,n}^2/2 + 1/8) > \epsilon) \\
&\le \mathbb{P}(n\exp(-(\min_{i \le n}\xi_{i,n}^2)/2 + 1/8) > \epsilon) \\
&\le \mathbb{P}(n\exp(-(\min_{i \le n} a_i^2/\hat{\sigma}_{n,-i}^2(x_i))/4 + 1/8) > \epsilon) \\
&\le \mathbb{P}(\exp(-a^2/(4\max_{i \le n}\hat{\sigma}_{n,-i}^2(x_i)) + 1/8 + \log n) > \epsilon)
\end{aligned}
$$

Since $\max_{i \le n} \hat{\sigma}_{n,-i}^2(x_i) = o_p(1/\log(n))$ by assumption, we further have $\mathbb{P}(\sum_i f(\xi_{i,n}) > \epsilon) \to 0$. Since $\epsilon > 0$ was arbitrary, we have shown that $\sum_{i=1}^n \hat{w}_i^n \xrightarrow{p} 0$.

Now, recall the form of the CLASH weighted z-statistic

$$\lambda_n^w = \frac{\sqrt{\sum_{1=1}^n \hat{w}_i^n}}{\sqrt{2\sigma^2}}\left(\frac{\sum_{i=1}^n \hat{w}_i^n y_i d_i}{\sum_{1=1}^n \hat{w}_i^n d_i} - \frac{\sum_{i=1}^n \hat{w}_i^n y_i(1-d_i)}{\sum_{1=1}^n \hat{w}_i^n(1-d_i)}\right)$$

Define $c$ such that $|y_i| \le c$. We know $c$ must exist, since the outcomes are bounded by assumption. Then, we have that

$$
\begin{aligned}
|\lambda_n^w| &= \left|\frac{\sqrt{\sum_{1=1}^n \hat{w}_i^n}}{\sqrt{2\sigma^2}}\left(\frac{\sum_{i=1}^n \hat{w}_i^n y_i d_i}{\sum_{1=1}^n \hat{w}_i^n d_i} - \frac{\sum_{i=1}^n \hat{w}_i^n y_i(1-d_i)}{\sum_{1=1}^n \hat{w}_i^n(1-d_i)}\right)\right| \\
&\le \frac{\sqrt{\sum_{1=1}^n \hat{w}_i^n}}{\sqrt{2\sigma^2}}\left(\frac{\sum_{i=1}^n \hat{w}_i^n c d_i}{\sum_{1=1}^n \hat{w}_i^n d_i} - \frac{\sum_{i=1}^n \hat{w}_i^n(-c)(1-d_i)}{\sum_{1=1}^n \hat{w}_i^n(1-d_i)}\right) \\
&= \frac{\sqrt{\sum_{1=1}^n \hat{w}_i^n}}{\sqrt{2\sigma^2}}2c \\
&\xrightarrow{p} 0
\end{aligned}
$$

Thus, we see that the weighted test statistic $\lambda_n^w$ converges in probability to 0. Now, the test can only reject if $\lambda_n^w$ exceeds a fixed and positive bound $b_\alpha$. By the definition of convergence in probability, this probability must shrink to zero.

## F   Additional Theoretical Results

**Corollary F.1** (CLASH weights converge quickly to optimal weights). *Assume that $\hat{\sigma}^2_{n,-i}(x_i) = o_p(1/\log(n))$. Then, Thm. 3.2 implies that $|\hat{w}^n_i - w^*_i| = o_p(1/n)$.*

*Proof.* Define $a_i = |\tau(x_i) - \delta|/\sqrt{2}$ and $Z_{i,n} = \frac{\tau(x_i) - \hat{\tau}_{n,-i}(x_i)}{\hat{\sigma}_{n,-i}(x_i)}$. Fix any $\epsilon > 0$, let $\xi_{i,n} = a_i/(\sqrt{2}\hat{\sigma}_{n,-i}(x_i))$, and define $f(x) = (1 + x/2)\exp(-x^2)$. Then, as in equation (7) in the proof of Thm. 3.2, we apply the law of total probability and equation (5) to yield

$$\mathbb{P}(n|\hat{w}^n_i - w^*_i| > \epsilon) \leq \mathbb{P}(|Z_{i,n}\hat{\sigma}_{n,-i}(x_i)| > a_i/(2\sqrt{2})) \quad +$$
$$\mathbb{P}\left(n\left(1 + \frac{a_i}{2\sqrt{2}\hat{\sigma}_{n,-i}(x_i)}\right)\exp\left(\frac{-a^2_i}{2\hat{\sigma}_{n,-i}(x_i)^2}\right) > \epsilon\right)$$
$$= \mathbb{P}(|Z_{i,n}\hat{\sigma}_{n,-i}(x_i)| > a_i/(2\sqrt{2})) + \mathbb{P}(nf(\xi_{i,n}) > \epsilon).$$

The first term on the right converges to 0 since $Z_{i,n}\hat{\sigma}_{n,-i}(x_i) = \tau(x_i) - \hat{\tau}_{n,-i}(x_i) \xrightarrow{p} 0$ (by the assumptions of Thm. 3.2). We focus on the second term. Since a convex function is no smaller than its tangent line, we have $(1 + \xi_{i,n}/2) \leq \exp(\xi_{i,n}/2)$ and hence

$$f(\xi_{i,n}) \leq \exp(-\xi^2_{i,n} + \xi_{i,n}/2) \leq \exp(-\xi^2_{i,n} + \xi^2_{i,n}/2 + 1/8) = \exp(-\xi^2_{i,n}/2 + 1/8)$$

where we used the arithmetic-geometric mean inequality in the final inequality. Thus,

$$\mathbb{P}(nf(\xi_{i,n}) > \epsilon) \leq \mathbb{P}(n\exp(-\xi^2_{i,n}/2 + 1/8) > \epsilon)$$
$$\leq \mathbb{P}(\exp(-a^2_i/4\hat{\sigma}^2_{n,-i} + 1/8 + \log n) > \epsilon)$$
$$\to 0,$$

since $\hat{\sigma}_{n,-i}(x_i)^2 = o_p(1/\log(n))$. Thus, $\mathbb{P}(n|\hat{w}^n_i - w^*_i| > \epsilon) \to 0$, and so by definition, $|\hat{w}^n_i - w^*_i| = o_p(1/n)$.

**Theorem F.2** (CLASH limits unnecessary stopping for the Gaussian SPRT). *Consider a stopping test with the Gaussian SPRT and weights estimated using CLASH. If $\max_{t \leq n} \hat{\sigma}_{n,-t}(x_t)^2 = o_p(1/\log(n))$, $\max_{t \leq n} |\tau(x_t) - \hat{\tau}_{n,-i}(x_t)| = o_p(1)$, and $z_t$ are uniformly bounded, then the stopping probability of the test converges to zero if no participant group is harmed.*

*Proof.* We adopt the setup for the SPRT described in App. B. Recall that the Gaussian SPRT test statistic is given by

$$\lambda_{2n} = \beta \sum_{t=1}^{n} z_t - \frac{n}{2}\beta^2$$

and the CLASH- weighted version of this statistic is given by

$$\lambda^w_{2n} = \beta \sum_{t=1}^{n} \hat{w}^n_t z_t - \frac{\sum_{t=1}^{n} \hat{w}^n_t}{2}\beta^2.$$

Now, by assumption, there exists some $c$ that bounds $|z_t|$ for all $t$. Since no participant is harmed, $w^*_i = 0$ for all $i$. Similar to the proof in App. E.3, if $\max_{i \leq n} \hat{\sigma}_{n,-i}(x_i)^2 = o_p(1/\log(n))$ and $\max_{i \leq n} |\tau(x_i) - \hat{\tau}_{n,-i}(x_i)| = o_p(1)$, then $\sum_{t=1}^{n} \hat{w}^n_t \to 0$. Thus, since $z_t$ are uniformly bounded by some $c$,

$$\lambda^w_{2n} \leq \beta \sum_{t=1}^{n} \hat{w}^n_t c$$
$$\xrightarrow{p} 0$$

This shows that $\lambda^w_{2n} \xrightarrow{p} 0$ as $n \to \infty$. Thus, $\mathbb{P}(\lambda^w_{2n} > \log \alpha) \to 0$ for any fixed $\alpha > 0$, proving the claim.

# G   Stopping Only on the Harmed Group

If the investigators choose to stop the experiment only on the harmed group, they face two practical challenges: identifying the harmed group and ATE estimation. We discuss each of these below.

**Identifying the harmed group.** To stop the experiment only for harmed participants, investigators must first identify the harmed group. This is non-trivial, since group membership is unobserved. The distribution of CLASH weights at stopping time can help in this task: groups with estimated weights close to 1 are likely to be harmed. Fig. S19 illustrates this in our empirical application: the estimated weights in Regions 1 and 2 are both close to 1, indicating that these are the groups on which to stop. With few covariates, investigators can manually inspect the weight distribution for each covariate combination to identify the harmed group. With many covariates, investigators can use a simple heuristic: a regression decision tree on the estimated CLASH weights can find the covariate values for which the weights are the largest. Limiting the depth of this tree can ensure that the identified group is actionable (i.e., it is possible to stop the experiment on it) and of non-trivial size.

There are alternative approaches to this harmed group identification task; for example, investigators can use subgroup identification methods on the raw outcomes (e.g. [1, 30, 44]). We are agnostic to the choice of method: investigators can pick the approach most appropriate for their domain.

**ATE estimation.** Stopping the experiment on only one group can affect inference at the end of the experiment, as the treatment is no longer randomly assigned across covariates. To estimate the whole population ATE, we recommend using inverse propensity weights (IPW) to correct for the induced selection bias. Let $\tilde{g}$ be the group that the experiment is stopped on and $p(\tilde{g})$ denote the proportion of the total population that comes from $\tilde{g}$. $p(\tilde{g})$ may either be known (e.g., from domain knowledge or prior experiments) or estimated from the interim data when the experiment is stopped on $\tilde{g}$. Let $p_{collected}(\tilde{g})$ denote the observed proportion of $\tilde{g}$ in the data collected over the entire experiment (i.e., pre- and post-stopping on $\tilde{g}$). Then, to estimate the whole-population ATE, observations from $\tilde{g}$ should be assigned a weight of $p(\tilde{g})/p_{collected}(\tilde{g})$ and all other observations should be assigned a weight of 1. This IPW approach will lead to unbiased ATE inference (see the example in Tab. S6).

# H    Additional Simulation Experiments

In this section, we present detailed results from our simulation experiments. App. H.1 presents results with Gaussian outcomes and App. H.2 presents results with time-to-event outcomes. In almost all settings, CLASH outperforms both the homogeneous and SUBTLE baselines. We note that CLASH's performance gains are relatively limited in three situations: (1) when the harmed group forms $50\%$ of the population (i.e., is no longer a true "minority") and the treatment has no effect on the remaining population, (2) when the experiment collects a very large number of covariates ($d = 500$), and (3) when the experiment recruits a very small number of participants ($N = 200$). However, even in these settings, CLASH's stopping probability is equivalent to the baselines.

## H.1    Gaussian Outcomes

### H.1.1    Setup

The simulation setup is the same as described in Sec. 4; we include it here again for easy reference. We consider a randomized experiment that evaluates the effect of $D$ on $Y$. Participants come from two groups, with $G \in \{0, 1\}$ indicating membership in a minority group. We do not observe $G$, but observe a set of binary covariates $X = [X_1, .., X_d]$, where $d$ varies between 3 and 10 and $p(X_j = 1) = 0.5 \; \forall \; j$. $X$ maps deterministically to $G$, with $G = \prod_{j=1}^{k} X_j$. We vary $k$ between 1 and 3: the expected size of the minority thus varies between 12.5% and 50% (of all participants). $Y$ is normally distributed, with $Y|G = 0 \sim \mathcal{N}(\theta_0 D, 1)$ and $Y|G = 1 \sim \mathcal{N}(\theta_1 D, 1)$. We vary $\theta_0$ between 0 and -0.1; the majority group is thus unaffected or benefited by the treatment. We vary $\theta_1$ between 0 and 1: the minority group is thus either unaffected or harmed.

The experiment runs for $N = 4000$ participants and $T = 2000$ time steps, recruiting one treated and one untreated participant at each step. The experiment has three checkpoints, with 1,000, 2,000, and 3,000 participants (corresponding to 25%, 50%, and 75% of the total time duration). At each checkpoint, we run CLASH and compute its stopping probability across 1,000 replications. In Stage 1, CLASH uses a causal forest with 5-fold CV and $\delta = 0.1$. Standard errors were estimated by the causal forest itself (i.e., not via the bootstrap). In Stage 2, CLASH uses one of four commonly-used stopping tests: an OF-adjusted z-test, an mSPRT [14], a MaxSPRT [16], and a Bayesian estimation-based test.

Each replication uses its own random seed. One replication—including all combinations of effect size, minority group size, number of covariates and stopping tests—takes between 58 and 82 minutes to run on a single CPU (depending on the random seed). This yields a total compute time of approximately 1,200 hours for the Gaussian experiments. The simulations were run in parallel on an academic computing cluster with over 200 CPUs. Note that each replication used one CPU with 4GB of RAM.

### H.1.2 Impact of Hyperparameter $\delta$

We first discuss the choice of hyperparameter $\delta$. We recommend that investigators set this value to the minimum effect size of interest (MESI). Here, we evaluate three different values of the MESI: 0.05, 0.1, and 0.2. Note that an effect size of 0.05 is detectable in this experiment (with $N = 4,000$) with 70% power; we thus do not consider MESI $< 0.05$, as such small effects would not be reliably detectable.

We present our results for CLASH with an OF-adjusted z-test and an mSPRT in Fig. S5. In most scenarios, all three settings of $\delta$ perform similarly. For the mSPRT with a 12.5% minority group, we note that a larger value of $\delta$ corresponds with higher stopping probability for larger effects. This is intuitive, as the higher $\delta$ is, the more weight CLASH will assign to parts of the covariate space that show clear signs of harm. However, CLASH performs better than the homogeneous and SUBTLE baselines for all three values of $\delta$. Thus, the results presented in the main text are robust to an increase or decrease in this hyperparameter.

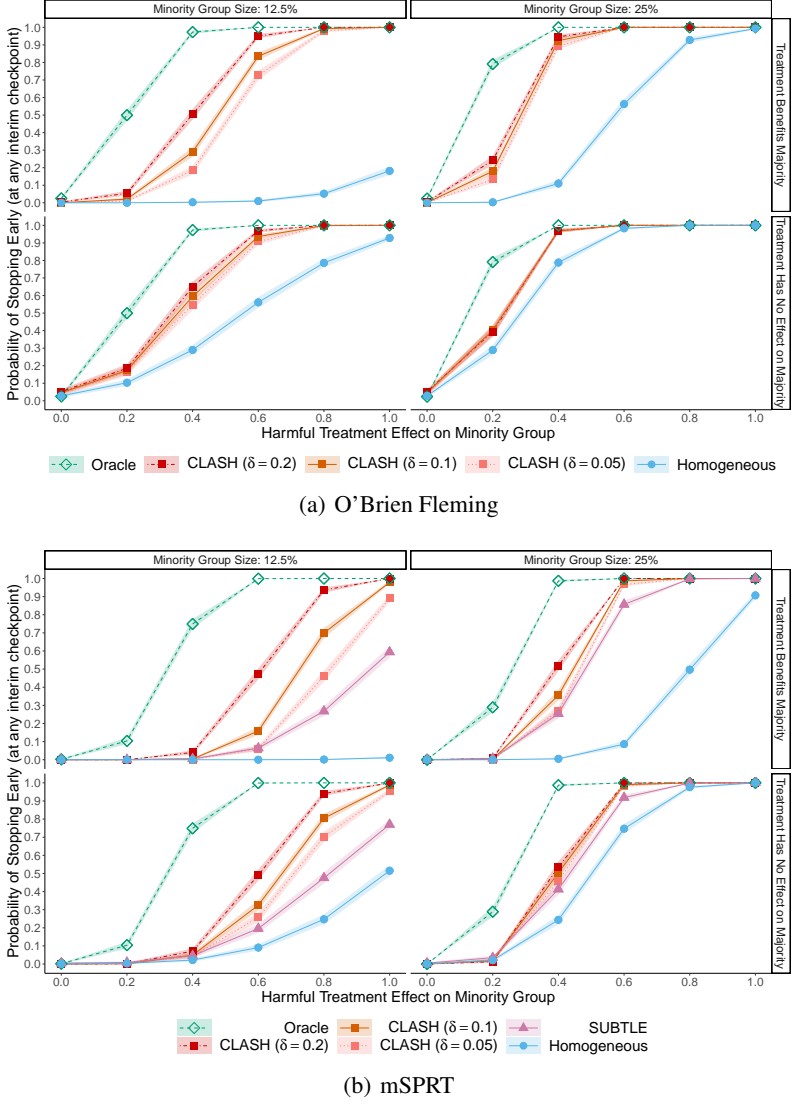

Figure S5: Effect of $\delta$ on CLASH's stopping probability. We consider the stopping probability of an (a) OF-adjusted z-test and (b) mSPRT across 1,000 replications of an experiment with Gaussian outcomes, $N = 4000$, and five covariates.

### H.1.3    Larger Minority Group

Fig. 2 in the main text summarizes CLASH's performance in an experiment where the minority group comprises 12.5% of the population. We now increase the size of the minority group, and present results in Fig. S6. With a 25% minority group, CLASH improves stopping probability over the homogeneous baseline, though the gains are more modest than for a 12.5% minority group. CLASH also performs slightly better than SUBTLE, though this difference is also reduced. All methods perform similarly with a 50% minority group.

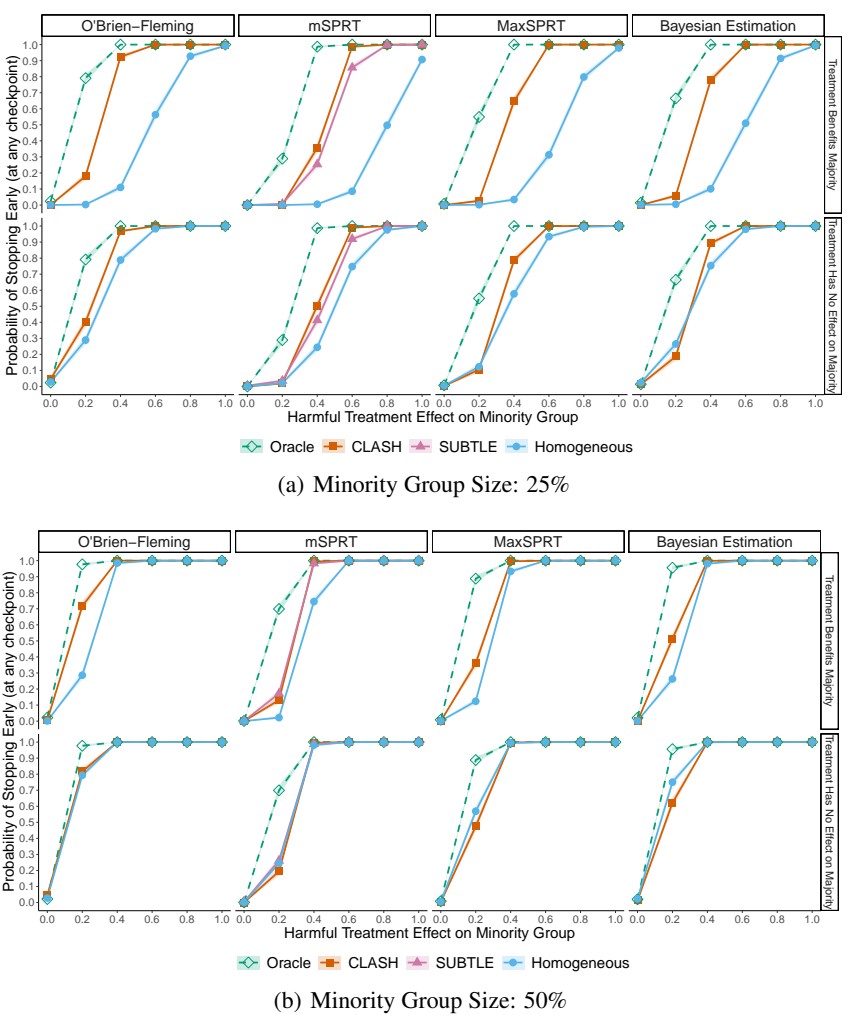

Figure S6: Performance of CLASH in a simulation experiment with Gaussian outcomes, five covariates, and a minority group that forms (a) 25% or (b) 50% of all participants. We simulate 1,000 trials with 4,000 participants each and plot the stopping probability (with 95% CI) at any interim checkpoint.

### H.1.4 Comparison with SUBTLE

Fig. 3 in the main text compares CLASH's performance to that of SUBTLE in when the treatment benefits the majority group and harms (or has no effect on) the minority. We present an analogous comparison in the case when the treatment has no effect on the majority group in Fig. S7. In most cases, CLASH increases the stopping probability over the SUBTLE baseline, especially at the first two interim checkpoints. The only exception is the setting with 10 covariates and a 12.5% minority group, when CLASH and SUBTLE perform similarly.

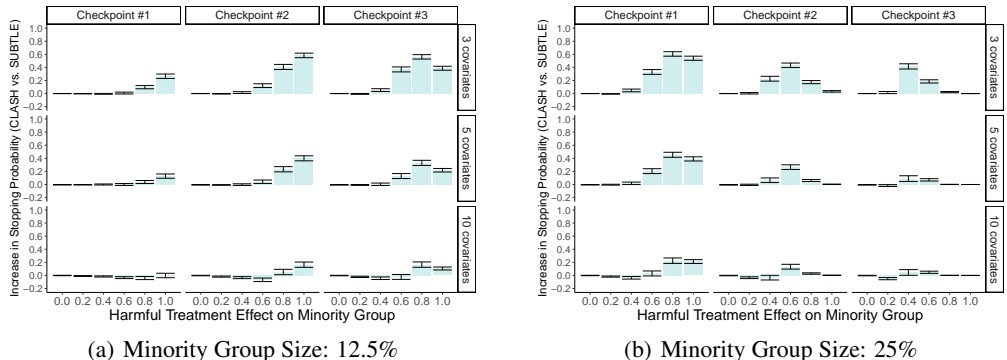

(a) Minority Group Size: 12.5%     (b) Minority Group Size: 25%

Figure S7: CLASH's performance improvement over the SUBTLE baseline with Gaussian outcomes, when the treatment has no effect on the majority group. We plot the difference in stopping probability between CLASH (with an mSPRT) and SUBTLE with 95% CIs, where the minority group forms (a) 12.5% or (b) 25% of all participants.

### H.1.5 Comparison with Homogeneous baseline

We now compare CLASH to the homogeneous baseline across a wide range of simulation settings. Fig. S8 compares the two methods when the treatment benefits the majority, while Fig. S9 compares them when the treatment has no effect on the majority. CLASH increases stopping probability in almost all cases.

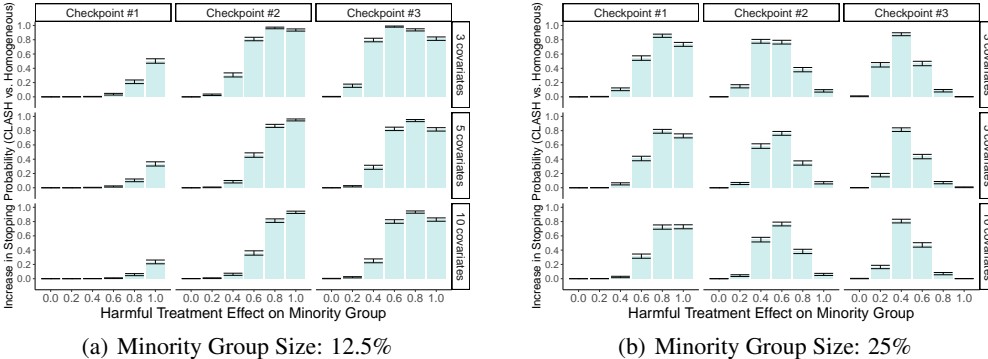

(a) Minority Group Size: 12.5%    (b) Minority Group Size: 25%

Figure S8: CLASH's performance improvement over the homogeneous baseline with Gaussian outcomes, when the treatment benefits the majority group. We plot the difference in stopping probability between CLASH (with an OF-adjusted z-test) and the homogeneous baseline with 95% CIs, where the minority group forms (a) 12.5% or (b) 25% of all participants.

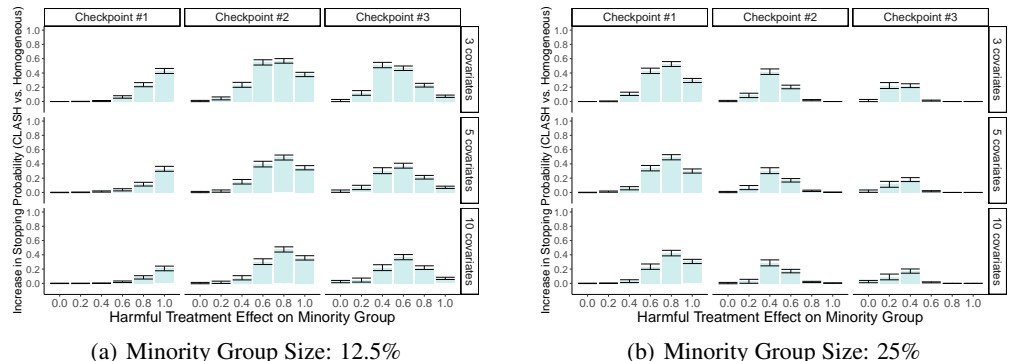

(a) Minority Group Size: 12.5%    (b) Minority Group Size: 25%

Figure S9: CLASH's performance improvement over the homogeneous baseline with Gaussian outcomes, when the treatment has no effect on the majority group. We plot the difference in stopping probability between CLASH (with an OF-adjusted z-test) and the homogeneous baseline with 95% CIs, where the minority group forms (a) 12.5% or (b) 25% of all participants.

### H.1.6 Small sample size

We evaluate CLASH's performance with much smaller sample sizes ($N = 200, 400, 1000$). We consider an experiment in which the treatment harms the minority group and weakly benefits the majority (i.e., $\theta_0 = -0.1$). There are five covariates and the minority group forms 12.5% of the population. Fig. S10 presents results for CLASH with an OF-adjusted z-test. CLASH outperforms the homogeneous baseline by a wide margin in experiments with moderately small samples (N=1000). While the gap between CLASH and the homogenous baseline decreases as sample size decreases, CLASH still outperforms the baseline with as few as 400 participants. However, with very small samples (N=200), neither CLASH nor the homogeneous baseline is likely to stop the experiment.

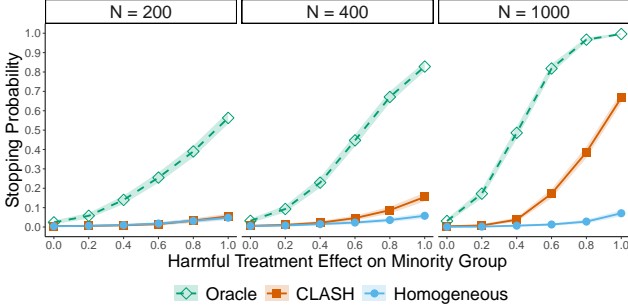

Figure S10: CLASH's performance in simulation experiments with small samples sizes (N=200, 400, 1000). We consider experiments with Gaussian outcomes, five covariates, a weakly benefitted majority group, and a 12.5% harmed minority group. We simulate 1,000 trials and plot the stopping probability (with 95% CI) at any interim checkpoint of CLASH (with an OF-adjusted z-test), the homogeneous baseline, and the Oracle.

### H.1.7 Large number of covariates

We evaluate CLASH's performance with high-dimensional covariate sets (100 and 500 covariates). We consider an experiment with 4,000 participants in which the treatment harms a 12.5% minority group and weakly benefits the majority (i.e., $\theta_0 = -0.1$). Fig. S11 presents results for CLASH with an OF-adjusted z-test. CLASH is robust to increasing dimensionality to a point, outperforming the homogeneous baseline even with 100 covariates. The extreme case with 500 covariates is more challenging: here, investigators may need to perform feature selection before running CLASH.

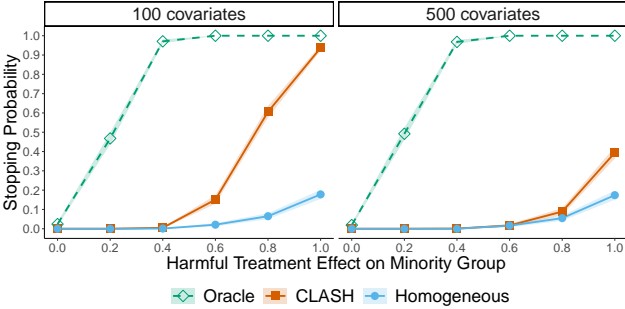

Figure S11: CLASH's performance in simulation experiments with high-dimensional covariate sets (100 and 500 covariates). We consider experiments with Gaussian outcomes, a weakly benefitted majority group, and a 12.5% harmed minority group. We simulate 1,000 trials with 4,000 participants each and plot the stopping probability (with 95% CI) at any interim checkpoint of CLASH (with an OF-adjusted z-test), the homogeneous baseline, and the Oracle.

### H.1.8 Multiple harmed groups

We evaluate CLASH's performance with multiple ($> 2$) groups in the population. We refer to the x-axis effect size values as $x$, and consider two settings:

a) Three groups of unequal size (group size and effect size, respectively, in parentheses): one weakly benefited (87.5%, $-0.1$), one strongly harmed (6.25%, $x$), and one weakly harmed (6.25%, $x/2$).

b) Four equally sized groups (effect sizes in parentheses): strongly benefited ($-x$), weakly benefited ($-x/2$), weakly harmed ($x/2$), and strongly harmed ($x$).

Fig. S12 displays the results. Overall, CLASH performs well: it stops more frequently than the homogeneous baseline across a range of effect sizes and as often as the Oracle for larger effects.

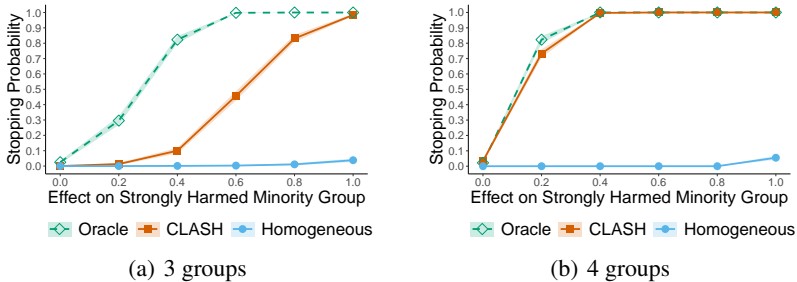

Figure S12: CLASH's performance in simulation experiments with multiple harmed groups. We consider experiments with Gaussian outcomes, five covariates, and (a) three groups (one weakly benefited majority group, one weakly harmed 6.25% minority group, one strongly harmed 6.25% minority group), or (b) four equally sized groups (one strongly benefited, one weakly benefited, one weakly harmed, one strongly harmed). We simulate 1,000 trials with 4,000 participants each and plot the stopping probability (with 95% CI) at any interim checkpoint of CLASH (with an OF-adjusted z-test), the homogeneous baseline, and the Oracle.

### H.1.9 Stochastic group membership

In previous experiments, the mapping from covariates to groups was deterministic. We now evaluate CLASH's performance in situations where the covariates map stochastically to the benefited and harmed groups. We construct a 25% minority group deterministically from covariates as before, but randomly assign $p\%$ of this group to be harmed. The remainder of the minority group and the majority group are both weakly benefited with effect size $\theta_0 = -0.1$. We consider experiments with $N = 4,000$ and five covariates, and plot results in Fig. S13. CLASH outperforms the homogeneous baseline both when $p = 0.5$ and $p = 0.75$.

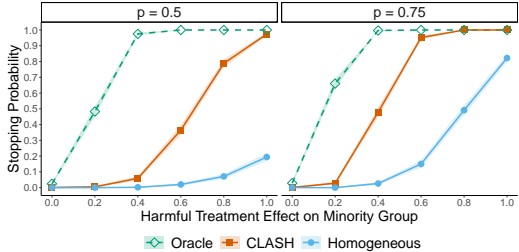

Figure S13: CLASH's performance in simulation experiments with stochastic group membership. We consider experiments with Gaussian outcomes, five covariates, and a weakly benefitted majority group. $p\%$ of participants from a 25% minority group are harmed by the treatment; the rest are weakly benefitted. We vary $p$ between 0.5 and 0.75. We simulate 1,000 trials with 4,000 participants each and plot the stopping probability (with 95% CI) at any interim checkpoint of CLASH (with an OF-adjusted z-test), the homogeneous baseline, and the Oracle.

### H.1.10 Smaller / larger outcome variance

We evaluate CLASH's performance as the variance in the outcomes increases. We consider an experiment with 4,000 participants and five covariates in which the treatment harms a 12.5% minority group and weakly benefits the majority ($\theta_0 = -0.1$). We vary $\mathrm{Var}(Y)$ between 0.5, 1, and 2 (recall that $\mathrm{Var}(Y) = 1$ in previous experiments). Fig. S14 demonstrates that CLASH outperforms the homogeneous baseline in all settings. CLASH and the Oracle both perform better with lower variance.

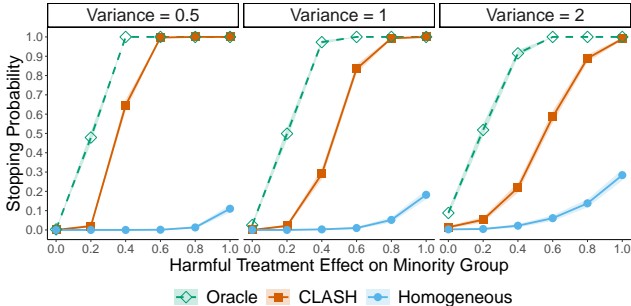

Figure S14: CLASH's performance in simulation experiments with smaller and larger variance in the observed outcomes. We consider experiments with Gaussian outcomes, five covariates, a weakly benefitted majority group, and a harmed 12.5% minority group. We simulate 1,000 trials with 4,000 participants each and plot the stopping probability (with 95% CI) at any interim checkpoint of CLASH (with an OF-adjusted z-test), the homogeneous baseline, and the Oracle.

### H.1.11 Unknown outcome variance

We now consider a setting in which the variance in the outcomes is unknown. The setup is exactly the same as in App. H.1.10, except that now we do not know $\mathrm{Var}(Y)$. Instead, we estimate the variance from the observed outcomes and use the plug-in version of the z-test. Fig. S15 demonstrates that while estimating the variance has a small effect on CLASH's performance, CLASH still drastically outperforms the homogeneous baseline.

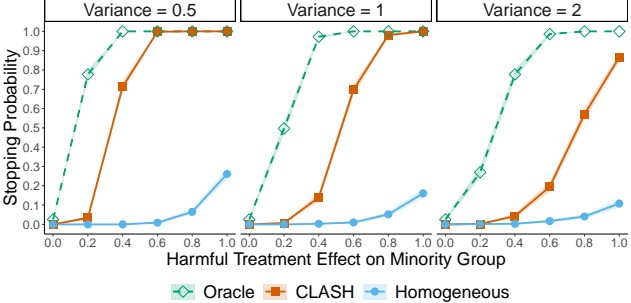

Figure S15: CLASH's performance in simulation experiments with unknown outcome variance. We consider experiments with Gaussian outcomes, five covariates, a weakly benefitted majority group, and a harmed 12.5% minority group. We simulate 1,000 trials with 4,000 participants each and plot the stopping probability (with 95% CI) at any interim checkpoint of CLASH (with an OF-adjusted z-test with plug-in variance), the homogeneous baseline, and the Oracle.

## H.2 TTE Outcomes

### H.2.1 Setup

We consider a clinical trial that measures time to a positive event (e.g., remission), and adapt the simulation setup from [20]. The trial runs for 30 months and recruits $2,000$ participants, who accrue uniformly at random over the study's first year. Some participants drop out of the trial: drop out time follows an exponential distribution with an annual hazard rate of 0.014. Group membership and covariates are generated as in the Gaussian case. The treatment effect is measured using the hazard ratio (HR), where $HR > 1$ is beneficial and $HR < 1$ is harmful. Outcomes follow a survival function with $S(t) = \exp\{-0.1(1 - D)t - \theta_0 D(1 - G)t - \theta_1 DGt\}$, with $\theta_1 \in [0.05, 0.1]$ and $\theta_0 \in \{0.1, 0.12\}$. The treatment thus either benefits or has non effect on the majority group, and either harms or has no effect on the minority group.

We conduct checkpoints at 12, 18, and 24 months.[8] CLASH's Stage 1 uses a survival causal forest [8] and $\delta = 0.05$. Stage 2 uses an OF-adjusted Cox proportional hazards regression. Note that SPRT-based tests have not yet been adapted to the TTE setting; among other reasons, the statistical dependence between checkpoints (induced by observing additional data from the same participant) proves challenging for these tests. Crucially, SUBTLE cannot be used either; CLASH is thus the first heterogeneous stopping method applicable to this setting. We compute CLASH's stopping probability at each checkpoint across 1,000 replications, and compare it to the homogeneous baseline and oracle.

As with the Gaussian experiments, each replication uses its own random seed. One replication— including all combinations of effect size, minority group size, number of covariates and stopping tests—takes between 137 and 202 minutes to run on a single CPU (depending on the random seed). This yields a total compute time of approximately 3,000 hours for the TTE experiments. The simulations were run in parallel on an academic computing cluster with over 200 CPUs. Note that each replication used one CPU with 4GB of RAM.

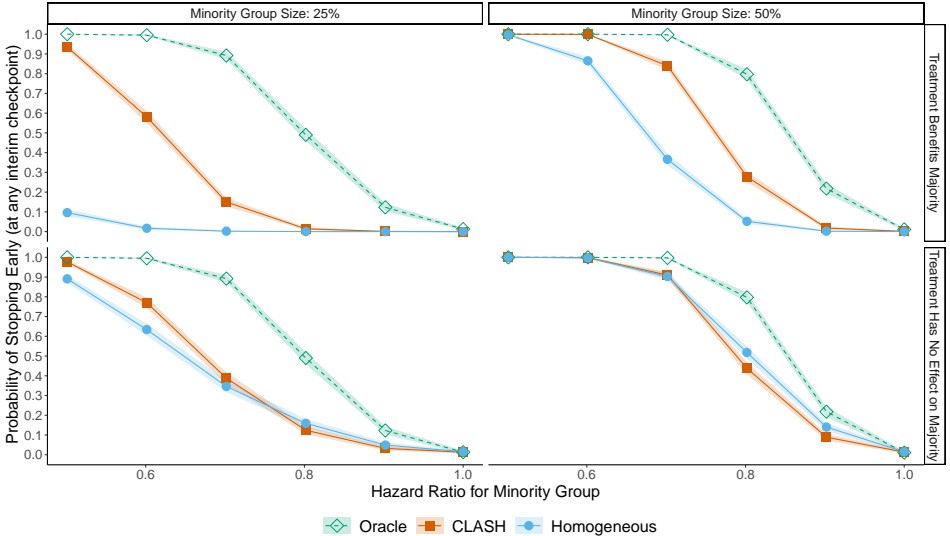

Figure S16: Performance of CLASH in a simulation experiment with TTE outcomes. If the minority group is harmed (lower hazard ratio), CLASH (red) significantly increases the stopping probability over the homogeneous approach (blue). This improvement is most notable when then treatment benefits the majority group. We simulate 1,000 trials with 2,000 participants and five covariates and plot the stopping probability (with 95% CI) at any interim checkpoint.

---

[8]We define checkpoints in terms of time, as TTE trials update observations for the same participants.

### H.2.2 Results

CLASH (with an OF-adjusted Cox regression) significantly improves stopping probability over the homogeneous baseline if the treatment benefits the majority group and harms the minority (Fig. S16). For a 25% minority group size, this performance improvement is most notable for large effect sizes ($HR \leq 0.6$), while for a 50% minority group size, the improvement is most notable for medium effects ($HR$ between 0.7 and 0.8). CLASH not only stops more often, but also faster, with higher stopping probability at earlier interim checkpoints (Fig. S17). These performance improvements are reduced when the treatment has no effect on the majority group (Fig. S18). Note that we do not consider harmed group sizes smaller than 25%, as no considered method was able to detect harmful effects with TTE outcomes on such small populations.

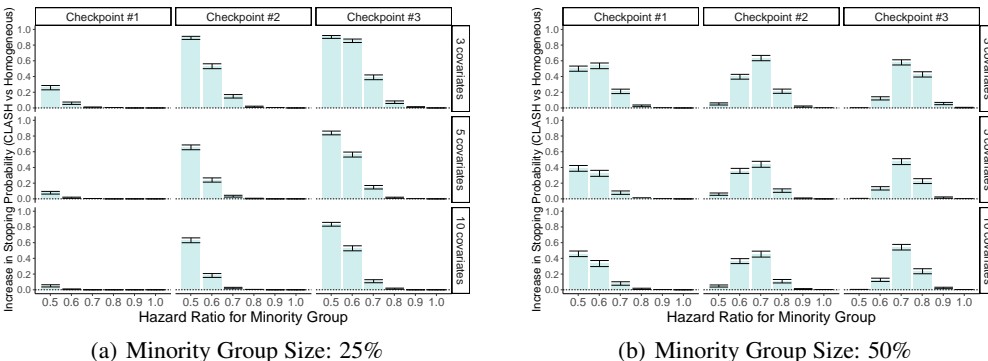

(a) Minority Group Size: 25%          (b) Minority Group Size: 50%

Figure S17: CLASH's performance improvement with TTE outcomes when the treatment benefits the majority group. CLASH significantly increases the stopping probability across a range of harmful effect sizes. CLASH not only stops more often, but also faster, boosting the stopping probability at earlier checkpoints. We plot the difference in stopping probability between CLASH and the homogeneous baseline (with 95% CIs), where the minority group forms (a) 25% or (b) 50% of all participants. The treatment effect is given by the hazard ratio (smaller HR indicates greater harm). For the larger group size, CLASH's improvement is most notable for medium effects, as the homogeneous baseline is also able to detect large effects. All performance boosts are robust to an increase in the number of covariates.

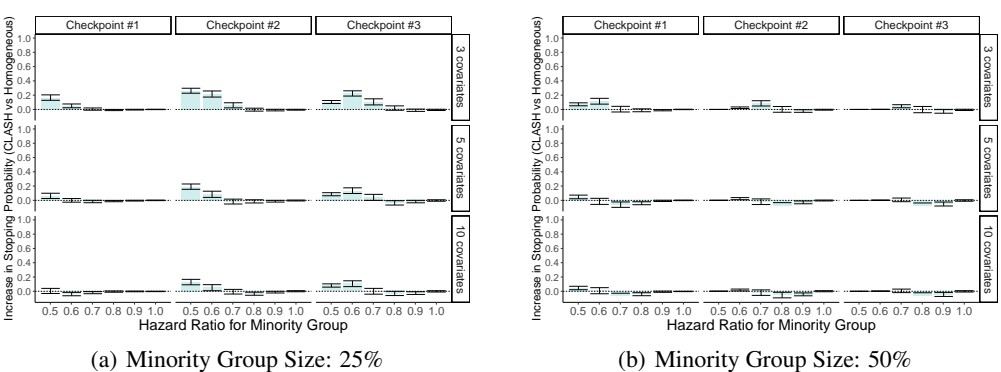

(a) Minority Group Size: 25%          (b) Minority Group Size: 50%

Figure S18: CLASH's performance improvement with TTE outcomes when the treatment has no effect on the majority group. We plot the difference in stopping probability between CLASH and the homogeneous baseline (with 95% CIs), where the minority group forms (a) 25% or (b) 50% of all participants. The treatment effect is given by the hazard ratio (smaller HR indicates greater harm).

# I   Real-world Application

We now present supplementary tables and figures that were referenced in our discussion of the real-world application in Sec. 5. We first present regional treatment effects, estimated with the whole dataset (Tab. S3) and at CLASH's stopping time (Tab. S4). We then present results from our semi-synthetic evaluation of CLASH's performance by sample size. In Fig. 4 in the main text, we only sampled from Region 1 (harmed group) and Regions 5-8 (unharmed group); this subset yielded a harmed group that comprised 29% of the total population. We now increase the size of the harmed group by including data from Regions 2-4. Note that we considered a region as harmed if the treatment effect estimated with the whole dataset (Tab. S3) was positive and significant; all other regions were considered unharmed. Further note that Fig. 4 in the main text only included Region 1 (instead of one of Regions 2, 3, or 4) since it was the most harmed region (Tab. S3). The results of this semi-synthetic evaluation (Fig. S20) are similar to those discussed in Sec. 5.

Table S3: Regional Effect of Treatment on Performance Metric (estimated at experiment's end with 500,000 observations). We present coefficients from separate univariate negative binomial regressions.

|          | Treatment Effect Estimate | Std. Error | p-value |
|----------|---------------------------|------------|---------|
| Region 1 | 0.385                     | 0.014      | 0.000   |
| Region 2 | 0.065                     | 0.016      | 0.000   |
| Region 3 | 0.107                     | 0.034      | 0.002   |
| Region 4 | 0.056                     | 0.010      | 0.000   |
| Region 5 | 0.006                     | 0.015      | 0.684   |
| Region 6 | 0.014                     | 0.017      | 0.424   |
| Region 7 | -0.048                    | 0.021      | 0.023   |
| Region 8 | -0.311                    | 0.025      | 0.000   |

Table S4: Regional Effect of Treatment on Performance Metric (estimated at checkpoint with 40,000 observations, i.e., when CLASH stops the experiment). We present coefficients from separate univariate negative binomial regressions.

|          | Treatment Effect Estimate | Std. Error | p-value |
|----------|---------------------------|------------|---------|
| Region 1 | 0.453                     | 0.051      | 0.000   |
| Region 2 | 0.376                     | 0.051      | 0.000   |
| Region 3 | 0.250                     | 0.130      | 0.054   |
| Region 4 | -0.146                    | 0.036      | 0.000   |
| Region 5 | -0.115                    | 0.065      | 0.078   |
| Region 6 | 0.260                     | 0.055      | 0.000   |
| Region 7 | -0.253                    | 0.077      | 0.001   |
| Region 8 | -1.292                    | 0.083      | 0.000   |

Table S5: Mean stopping time in our empirical application. We shuffle our real-world dataset 1,000 times and compute the mean stopping time (with standard errors) for CLASH, the homogeneous baseline, and the Oracle.

| Method      | Mean Stopping Time (Std. Error) |
|-------------|---------------------------------|
| Oracle      | 48,500 (431)                    |
| CLASH       | 57,200 (609)                    |
| Homogeneous | 64,420 (848)                    |

Table S6: ATE estimation in our empirical application if the experiment is only stopped for the harmed group. We assume that the experiment is stopped in Region 1 at CLASH stopping time of 40,000 participants and continued in all other regions. We consider the ATE without stopping (i.e., after continuing to collect data from all regions) as the ground truth. The naive ATE estimate (i.e., an unweighted estimate using all collected data) is biased and underestimates the true ATE. Using the IPW approach detailed in App. G corrects this bias.

|  | Estimate (Std. Error) |
| --- | --- |
| ATE estimate without stopping | 0.11 (0.006) |
| Naive ATE estimate with stopping | 0.03 (0.006) |
| IPW ATE estimate with stopping | 0.10 (0.006) |

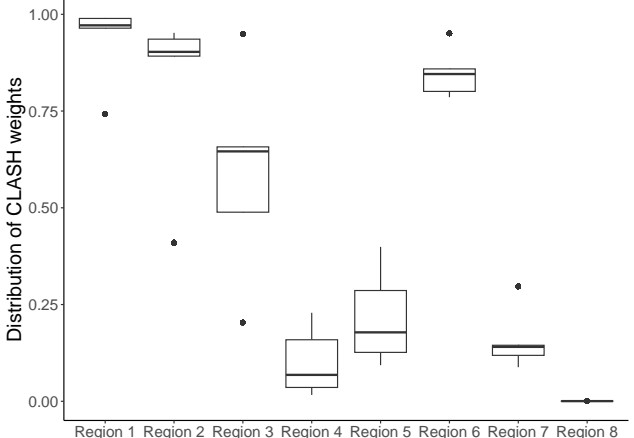

Figure S19: Distribution of CLASH-estimated weights by region (at checkpoint with 40,000 observations, i.e., when CLASH stops the experiment). CLASH identifies that the treatment has a strong harmful effect in Regions 1 and 2 and assigns these observations unit weight.

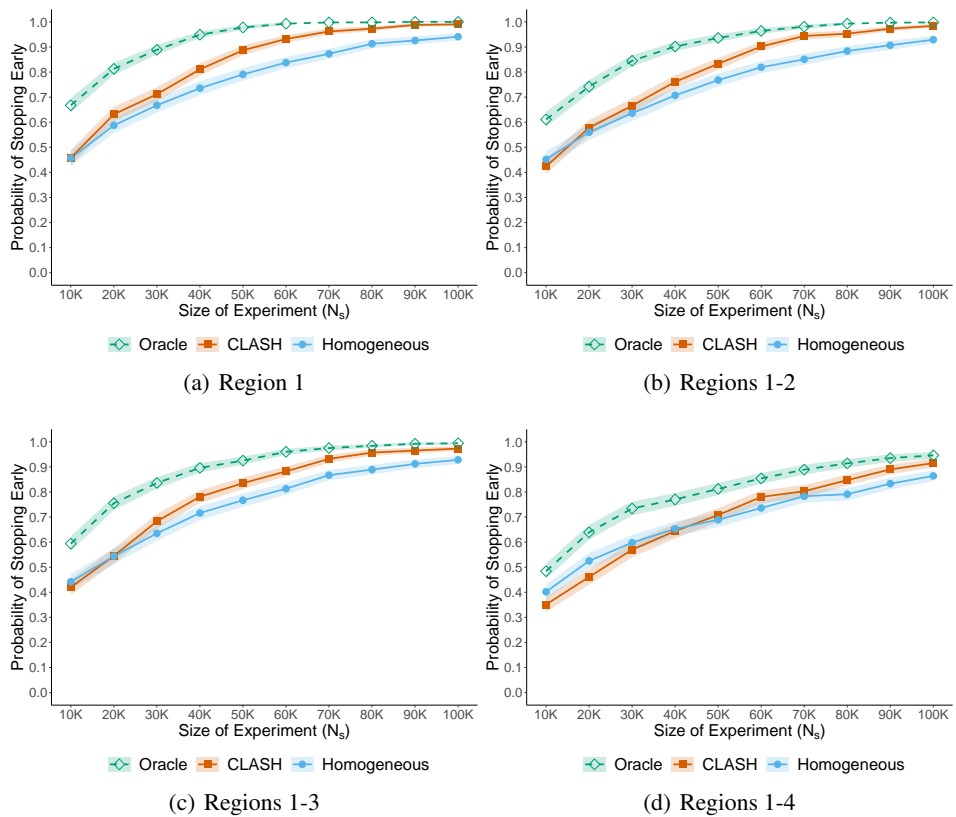

Figure S20: CLASH's performance by sample size with real data from an A/B test. We plot stopping probability across 1,000 random samples with 95% CIs. We define the unharmed group as Regions 5-8. We define the harmed group as (a) Region 1 (29% harmed group size), (b) Regions 1-2 (41% harmed group size), (c) Regions 1-3 (43% harmed group size), or (d) Regions 1-4 (60% harmed group size). In cases (a) - (c), CLASH significantly increases stopping probability over the homogeneous baseline for all $N_s \geq 50,000$, and achieves near-oracle level performance with $N_s = 100,000$ (20% of the overall experiment's size). Case (d) reflects a situation in which a majority group is harmed, which is not our method's primary focus. However, CLASH still outperforms the homogeneous baseline by $N_s = 100,000$.

