# OpenReview forum: "Should I Stop or Should I Go: Early Stopping with Heterogeneous Populations"
_NeurIPS.cc/2023/Conference — NeurIPS 2023 spotlight_

### Official Review · Reviewer_tX2R · 2023-07-01

**Soundness:** 3 good
**Presentation:** 4 excellent
**Contribution:** 3 good
**Rating:** 6
**Confidence:** 4

**Summary:**

This work proposes a method for adapting stopping tests of randomized experiments in heterogeneous populations. Specifically, the authors motivate the problem, namely why heterogeneous treatment effects lead to late stopping of randomized experiments, for instance when a minority group is harmed. They then propose a two stage method which first predicts a weighting of the original test statistic components used in stopping decisions, then and then uses these statistics to make the stopping decision. The methodological contribution is well-motivated and supported by theoretical results which analyse the convergence behavior of these weights, and the probability of stopping under the assumption of knowing the group membership.

**Strengths:**

* There is a lack of machine learning methods that addresses the heterogeneous early stopping problem, and this paper provides a possible, first solution while making few assumptions. This renders the work an original, well-motivated contribution.
* The paper is very clear and well written. In particular, the links between each of the sections are very clear.
* The experimental results, including the simulated scenarios, are convincing and interesting. For instance Figure 2 makes clear why the proposed approach is advantageous over homogeneous stopping tests.


**Weaknesses:**

* The task is somewhat niche. It is furthermore unclear to what degree the stopping task in randomized experiments could be reformulated as a similar task in another domain, i.e. to what degree this or similar problems have been solved in other contexts.
* Overall, the work makes various idealised assumptions in both the theoretical results, and in the (synthetic) experiments considered. It is an interesting proof of concept, but there is a lot of work which would need to be done to make this method applicable in practical settings, for instance real-world clinical trials which this method is motivated by, but does not evaluate on. Two further points to note is the lack of performance on high-dimensional data that the authors themselves notes, which is present in some clnical trials. Second, the method crucially relies on treatment effect estimation methods in Stage 1, which themselves are far from being widely applicable in practice.


**Questions:**

* The problem setup of CLASH stops the entire experiment if a stopping decision has been made. However, in clinical trials for example, it may not be ethical to stop the experiment for a majority group which benefits from the treatment. Could CLASH be adapted to an online setting where the experiment is only stopped for the harmed subgroup? It would be interesting if the authors could comment on this.

**Limitations:**

* Prop. 3.1 assumes the group membership (and CATE) is known, which is never the case in practice. The authors state this, yet it is unclear how one would efficiently infer group membership in large-scale scenarios in practice, and whether the proposed solution in Stage 1 of the algorithm would work. It is consequently unclear how well the proposed tests would generalise if group membership is not known, or how the asymptotic behavior analysed in this Proposition would change in this case. It would be helpful if the authors could comment on this.
* The experimental settings are all limited to two groups. In many real-world settings such as clinical trials, we would expect more than two groups with somewhat homogeneous treatment effect. How this method would perform in such cases is unclear, and neither discussed in the paper.

---

> ### Author Rebuttal · Authors · 2023-08-09
>
> Thank you for your detailed review: we appreciate your positive feedback and constructive suggestions. We address each of your comments in further detail below.
>
> **R6: _Overall, the work makes various idealised assumptions in both the theoretical results, and in the (synthetic) experiments considered. It is an interesting proof of concept, but there is a lot of work which would need to be done to make this method applicable in practical settings, for instance real-world clinical trials which this method is motivated by, but does not evaluate on. Two further points to note is the lack of performance on high-dimensional data that the authors themselves notes, which is present in some clinical trials. Second, the method crucially relies on treatment effect estimation methods in Stage 1, which themselves are far from being widely applicable in practice._**
>
> Thank you for raising this important point. Our global response considers several additional simulation settings, including on high-dimensional data (Fig 2e in the global response). We find that CLASH is able to perform well even in experiments with 100 covariates. Overall, we believe the additional simulations indicate the CLASH can be effective in situations closer to real world clinical trials and A/B tests.
>
> Further, we emphasize that CLASH does not require using machine learning-based methods for causal estimation. Practitioners can use much simpler techniques–including linear regression–to infer heterogeneous treatment effects in stage 1. CLASH is agnostic to the specific method used: practitioners can use the method with which they are most comfortable, as long as it yields reasonably accurate estimates of the effect heterogeneity.
>
> **R6: _The problem setup of CLASH stops the entire experiment if a stopping decision has been made. However, in clinical trials for example, it may not be ethical to stop the experiment for a majority group which benefits from the treatment. Could CLASH be adapted to an online setting where the experiment is only stopped for the harmed subgroup? It would be interesting if the authors could comment on this._**
>
> This is an excellent point: we discuss this in detail in point (2) of our global response and have updated our manuscript accordingly to address this decision-making process. In short, yes – CLASH can be used to inform early stopping on only the harmed subgroup rather than the entire trial population.
>
> **R6: _Prop. 3.1 assumes the group membership (and CATE) is known, which is never the case in practice. The authors state this, yet it is unclear how one would efficiently infer group membership in large-scale scenarios in practice, and whether the proposed solution in Stage 1 of the algorithm would work. It is consequently unclear how well the proposed tests would generalise if group membership is not known, or how the asymptotic behavior analysed in this Proposition would change in this case. It would be helpful if the authors could comment on this._**
>
> Thank you for giving us the chance to clarify: group membership knowledge is not needed when using CLASH. You are correct to point out that the test in Prop 3.1 could never be used in practice, as it requires prior knowledge of the groups and treatment effects. However, it is an important result because it describes the statistical power of the test that CLASH converges to in large samples. Thm 3.2 establishes that in large samples, CLASH converges to the test described in Prop 3.1. Prop 3.1 indicates that this test has power-1 in large samples; thus CLASH must also have power-1 in large samples. We emphasize that CLASH does not require knowledge of group membership to operate: it infers this in stage 1 using the provided covariates and observed outcomes. Thus, CLASH is able to obtain the optimal power of the test described in Prop 3.1 despite not knowing group membership a priori. Our simulation experiments and empirical application demonstrate that CLASH efficiently infers group membership in practice.
>
> **R6: _The experimental settings are all limited to two groups. In many real-world settings such as clinical trials, we would expect more than two groups with somewhat homogeneous treatment effect. How this method would perform in such cases is unclear, and neither discussed in the paper._**
>
> Thank you for raising this important concern. Figure 1 in our global response considers situations with more than two groups, and demonstrates that CLASH can be effective in such settings.

---

> > ### Comment · Reviewer_tX2R · 2023-08-10
> > **Response to rebuttal**
> >
> > My points of concern are largely resolved, my questions answered. I have also read through the other reviews and their responses.
> >
> > I think this paper particularly shines in tackling an interesting, understudied problem, for which no good solutions seem to exist. It is simple and straight-forward, yet solid. I vote for accepting the work.

---

### Official Review · Reviewer_R8bL · 2023-07-05

**Soundness:** 3 good
**Presentation:** 4 excellent
**Contribution:** 3 good
**Rating:** 7
**Confidence:** 3

**Summary:**

The paper focuses on stopping tests for harm in clinical trials or A/B testing where heterogeneous treatment effect is involved. The proposed method contains two phases: First, the population harmed by the trials is identified via conditional treatment effect estimation. Second, weighted versions of widely-used test statistics are computed to determine if the trial needs to be stopped early. Theoretical analyses are carried out to show that the proposed method meets the desired requirements: producing high probability when a subgroup is harmed while limiting the unnecessary stopping. Experiments using simulation and real-world data show improvement over homogeneous stopping tests and the existing heterogeneous ones.

**Strengths:**

+ The paper is well-written and easy to follow. The research problem is clearly defined and the motivation for the proposed method is nicely presented.
+ Theoretical analyses show that the proposed method could achieve the desired properties. The convergence property (Thm. 3.2.) is particularly important and well-explained.

**Weaknesses:**

- In lines 191 to 193, $(x_i, y_i)$ is excluded from training set when estimating $\tau(x_i)$. Does it mean that the CATE estimation model needs to be trained for $n$ times during stage 1? Are there any comparisons of the running time between baseline models, say SUBTLE?

**Questions:**

- In the lower part of Fig. 2, when the treatment has no effect on the majority, why CLASH has lower probabilities of stopping early than the homogeneous approach when the harmful treatment effect on the minority group is below 0.4?

**Limitations:**

The limitations of the proposed method are discussed in the last section. It would be great if the authors could add a few sentences about how they plan to address the limitations in future work.

---

> ### Author Rebuttal · Authors · 2023-08-09
>
> Thank you for your detailed review: we appreciate your positive feedback and constructive suggestions. We address each of your comments in further detail below.
>
> **R5: _In lines 191 to 193, $(x_i,y_i)$ is excluded from training set when estimating $\tau(x_i)$ Does it mean that the CATE estimation model needs to be trained for n times during stage 1? Are there any comparisons of the running time between baseline models, say SUBTLE?_**
>
> Our apologies for not being more clear. While it is possible to use leave-one out cross-validation---in which case the model would need to be trained n times---we recommend using k-fold cross-validation or progressive cross-validation instead. In this case, the model would only need to be trained k times during stage 1. Our simulation experiments and empirical application use 5-fold cross-validation; we will make this more clear in the revised manuscript. In our empirical application, at an interim checkpoint with 40,000 participants (i.e., when CLASH indicates that the experiment should be stopped), all four methods take under a minute to run, though CLASH is the slowest (clocking in at 36 seconds).
>
> **R5: _In the lower part of Fig. 2, when the treatment has no effect on the majority, why CLASH has lower probabilities of stopping early than the homogeneous approach when the harmful treatment effect on the minority group is below 0.4?_**
>
> Thank you for raising this important point. Overall, the situation in which the majority group is unaffected and the minority group is only slightly harmed reflects both an easy case for the homogeneous baseline and a difficult case for CLASH. CLASH performs better when it is easy to differentiate the harmed and unharmed groups based on the covariates and outcomes. The more similar the effects on the minority and majority groups, the harder this task is. Meanwhile, the homogeneous baseline is the exact opposite: it performs better when the whole population ATE is more similar to the effect on the minority group. However, we emphasize that even in this case, the difference between the two methods is small and only exists for the Bayesian estimation-based stopping test.

---

> > ### Comment · Reviewer_R8bL · 2023-08-18
> > **My concerns have been addressed**
> >
> > Thank the authors for the response. I have read the responses, as well as the discussion between the authors and other reviewers. I think my main concerns have been clarified/addressed. Overall, this is a solid paper, so I am happy to raise the rating to 7.

---

### Official Review · Reviewer_SD94 · 2023-07-17

**Soundness:** 3 good
**Presentation:** 3 good
**Contribution:** 3 good
**Rating:** 6
**Confidence:** 3

**Summary:**

The authors propose an approach to early stop clinical trials in order to prevent subgroup level harms. Their approach involves first estimation of sequential estimation of a an individualized treatment effect using machine learning methods followed by reweighting the test statistic at each iteration with the estimated mean and standard deviation of the CATE.

Furthermore their choices are such that they do not need to make apriori assumptions on what groups represent the minority subgroups.

The authors present results around optimality of the metrics using the estimated CATE and perform extensive real world and synthetic experiments to demonstrate the effectiveness of their approach.

**Strengths:**

* The problem of early stopping of a clinical trial to prevent aggregate and subgroup level harms is an important one. The contribution is timely and relevant.

* The demonstration of the proposed method on the time-to-event (survival) setting is welcome since outcomes in most real world clinical trials are censored time-to-events.

* The extensive theoretical insights and real world and synthetic experiments are thoughful and welcome.

**Weaknesses:**

* The paper requires estimates of the uncertainity in CATE at an *individual* level. CATE is itself hard to estimate, and its uncertainity is even harder in practice to estimate. The authors propose to use "bootstrap" to compute this quantity. This seems practically impossible especially with high dimensional covariates.

*  While the paper is motivated strongly there is a fundamental question that remains to be addressed: The current method is such that it does not require apriori assumptions on which covariates specify the subgroups. It is unclear as to how in practice the group would be specified implicitly by the model. For instance, in the case of a large RCT there would always exist trivially small subgroups that are harmed. However these subgroups would not be generalizable and such results would not transport. Ideally this should be reflected in the uncertainity estimates upto generalization error. However this maybe violated in small sample size regimes.




**Questions:**

* The cdf for chosen weights, $w$ corresponds to a normal distribution. I think this is because the estimated weights tend to a normal distribution by central limit theorem, but I was unable to find a result for this in the paper can the authors point to this.

* Can the authors address the second weakness pointed out in more details. Specifically does the current setup allow a practioner to specify something along the lines of a "minimum" group size to prevent the model from raising false alarms with trivial subgroups?

* Finally, the current setup it seems does not allow for anyway of specifying how the covariates across different subgroups are related. For a subgroup to be actionable there should be similarities in there covariates. Is there a way to enforce or obtain such similarities from the experiment.

Overall I am willing to readjust my scores favorably based on answers to my questions as well as discussions and deliberations during the author rebuttal phase.

**Limitations:**

The paper does raise an interesting concern over whether current trials should be stopped for the entire population if a single subgroup is found to be harmed from the intervention. As the authors point out, this is a question of medical ethics and is largely beyond the scope of this manuscript, and hence I am inclined to not consider it as a "limitation". I think however that perhaps some examples of where such decisions can lead to different outcomes should be included in the manuscript.

---

> ### Author Rebuttal · Authors · 2023-08-09
>
> Thank you for your detailed review: we appreciate your positive feedback and constructive suggestions. We discuss each of your comments in detail below, and incorporate our responses in the revised paper.
>
> **R4: _The authors propose to use "bootstrap" to compute this quantity. This seems practically impossible especially with high dimensional covariates._**
>
> Thank you for allowing us to clarify. You are correct: CLASH requires estimating uncertainty in individual CATE estimates. However, the bootstrap is merely one way to obtain such uncertainty estimates. For example, a causal forest estimates these uncertainties in the process of fitting, and thus requires little additional computation. We recommend using such methods with CLASH wherever possible. Bootstrapping is a last-resort in cases where there are strong domain-specific reasons to use a certain CATE estimation method, and that method is unable to estimate the required uncertainties on its own. We have added this suggestion to the methods section of our revised paper.
>
> **R4: _While the paper is motivated strongly there is a fundamental question that remains to be addressed…it is unclear as to how in practice the group would be specified implicitly by the model. For instance, in the case of a large RCT there would always exist trivially small subgroups that are harmed. However these subgroups would not be generalizable and such results would not transport._**
>
> Thanks for raising this important point. As detailed in Part 2 of our global response, when CLASH indicates that an experiment should be stopped, practitioners can identify the specific harmed groups either by analyzing the distribution of the estimated CLASH weights or by using existing subgroup identification methods (see global response for more details). We also suggest ways to ensure that the identified harm subgroups are not trivially sized (e.g., limiting depth for tree-based methods). In general, domain expertise must play an important role in determining what is and what is not a meaningful group on which to stop.
>
> Regarding inaccurate uncertainty estimates in small samples: in Part 1 of the global response, we find that CLASH rarely stops experiments with very low N (Fig 2d). Thus, CLASH displays the opposite problem with small samples: it does not stop experiments due to trivially-sized harmed groups, but rather is unable to detect harmful effects with low N (note that the homogenous baseline is also unable to detect such effects). In general, we advise using CLASH with caution in experiments with small samples; we discuss this further in our limitations section.
>
> **R4: _The cdf for chosen weights corresponds to a normal distribution. I think this is because the estimated weights tend to a normal distribution by central limit theorem, but I was unable to find a result for this in the paper can the authors point to this._**
>
> Thank you for giving us the opportunity to clarify. The normal CDF is used primarily because it leads to fast convergence of the CLASH weights to the optimal weights (see Thm 3.2), not because the weights have an asymptotically normal distribution. It may be possible to use other CDFs and achieve similar results; however, the normal CDF proved to be an effective choice, as it yields provably fast weight convergence. We emphasize that CLASH does not require any central limit theorem-like assumptions on either the weights or the estimated CATEs.
>
> **R4: _Can the authors address the second weakness pointed out in more details. Specifically does the current setup allow a practioner to specify something along the lines of a "minimum" group size to prevent the model from raising false alarms with trivial subgroups?_**
>
> Per Part 2 of our global response, we recommend a tree-based heuristic that practitioners can use to identify the harmed groups. If this heuristic implies that only a trivially sized group is harmed, practitioners can ignore CLASH’s recommendation and continue the experiment. However, this decision depends heavily on the ethical and financial considerations of the practitioners.
>
> **R4: _...For a subgroup to be actionable there should be similarities in there covariates. Is there a way to enforce or obtain such similarities from the experiment._**
>
> Thank you for raising this concern. We agree that to stop an experiment only on one group, the group should have similar values for a few key covariates. This relates closely to our discussion above on identifying the harmed group (further detailed in point 2 of our global response). Practitioners can identify such harmed groups in an actionable way by using either our tree-based heuristic or existing subgroup identification techniques (e.g. [1]). Limiting the depth of the tree-based heuristic offers an easy way to ensure the identified group is actionable and of non-trivial size. For example, by limiting the tree-depth to 2, investigators can identify the two covariates that most drive harm and get a well-defined group on which to stop the experiment (if this is what they choose to do). Subgroup identification techniques provide analogous ways to get actionable groups (e.g. limiting tree-depth, variable selection, etc.).
>
> [1] Zhang, et al, DOI: 10.21037/atm.2018.03.07
>
> **R4: _Limitation: …perhaps some examples of where such decisions can lead to different outcomes should be included in the manuscript._**
>
> We address this limitation in further detail in part (2) of our global response, including examples from both clinical and technology domains of whether / when to stop on a subgroup; we will also provide these examples in the revised paper.
>
> **R4: _Overall I am willing to readjust my scores favorably based on answers to my questions as well as discussions and deliberations during the author rebuttal phase._**
>
> Thank you for your detailed feedback. We hope our response has helped answer your comments, and look forward to further discussion this week.

---

### Official Review · Reviewer_K2p1 · 2023-07-18

**Soundness:** 3 good
**Presentation:** 4 excellent
**Contribution:** 3 good
**Rating:** 7
**Confidence:** 3

**Summary:**

The authors propose CLASH, a method for early stopping in RCTs and A/B tests on heterogeneous populations. They some theoretical results that show that CLASH works, and provide simulations and one real experiment.

**Strengths:**

- Clear writing
- Good motivation
- Excellent exposition of theoretical results for readers who may not be able to fully understand proof details
- Convincing (albeit limited) experimental evidence

**Weaknesses:**

The biggest issue with this paper has to do with the experiments.

EDIT: I have raised my score to a 7 after the authors answered my questions.

- Unrealistic settings in the simulation - see the questions section.
- Your real-world data experiment is nice, but it's a single experiment with a single outcome - you could be doing as well as Oracle purely by chance. You could run it multiple times by randomizing the order of the data and reporting a distribution of stopping times against oracle, SUBTLE, and homogeneous.
- "Note that stopping the experiment just in one region would affect statistical inference at the end of the experiment, as the treatment would no longer be randomly assigned across regions. Practitioners can use covariate adjustment, inverse probability weighting, or adaptive sampling methods [13] to adjust for this selection." I would recommend actually performing this calculation and reporting the outcome.


**Questions:**

- In the simulation you have "X maps deterministically to G" -> is this not unrealistically simple? What happens if you use a stochastic function with varying degrees of noise?
- Similarly, "recruiting one treated and one untreated participant at each step" is unrealistic. Why not try to mimic real recruitment procedures and have batch recruitment? Does this affect your results at all?
- In your simulation "Y is normally distributed" with a fixed standard deviation of 1.0. What happens if the standard deviation is larger or smaller? Is it important to know this standard deviation in practice?
- You write "All performance increases are robust to an increase in the number of covariates" but also "CLASH works better with a relatively small number of covariates": these two statements conflict, and I did not see any experiments that demonstrate the latter. It's important to add experiments that quantify how your models perform (or don't) in higher dimensions. What happens when the dimensionality is 100 and 1000 in the simulation?
- "We only sample from Region and Regions 5-8; this gives us one harmed group (Region 1) that comprises 28% of the total population" - why do this? Why not sample uniformly at random from the entire dataset?

**Limitations:**

Yes

---

> ### Author Rebuttal · Authors · 2023-08-09
>
> Thank you for your detailed review: we greatly appreciate your insightful feedback. We address each of your constructive suggestions below, which we believe have further strengthened the paper.
>
> **R3: _The biggest issue with this paper has to do with the experiments. I'll happily raise my score to a 7 once additional experimental results are included…_**
>
> Our global response presents results from additional simulation experiments, many of which directly address your comments below. We include these results in the appendix of the revised paper.
>
> **R3: _Your real-world data experiment is nice…you could run it multiple times by randomizing the order…_**
>
> Thank you for raising this important point. We have included your suggested analysis in the revised paper. We find that CLASH stops the experiment at the same interim checkpoint as the Oracle in 62.6% of shuffled datasets. The mean (std. error) stopping times for each method across 1,000 shuffles is below.
>
> CLASH: 57,200 (609)
>
> Homogeneous: 64,420 (848)
>
> Oracle: 48,500 (431)
>
> **R3: _‘Stopping the experiment just in one region would affect statistical inference’... I would recommend...performing this calculation…_**
>
> Thank you for this suggestion. We report results from our empirical application below and in the revised paper. We assume practitioners choose to stop the experiment in the region with the largest harmful effect (Region 1). This choice leads to bias in the naive estimate of the full population ATE; however, inverse propensity weighting (IPW) corrects the bias and successfully recovers the ATE.
>
> ATE estimate without stopping: 0.11 (0.006)
>
> Naive ATE estimate with stopping: 0.03 (0.006)
>
> Estimate using IPW: 0.10 (0.006)
>
> **R3: _X maps deterministically to G -> is this not unrealistically simple? What happens if you use a stochastic function...?_**
>
> Your comment is well-taken, especially for clinical trials. To address your question, Fig 2a in the global response demonstrates that CLASH still performs well even with stochastically determined groups. That said, we would like to note that deterministic mappings from covariates to harmed groups do occur in real-word settings. For example, a new product feature evaluated in an A/B test may increase system crashes for only certain device types (e.g., Android devices with a certain chip). In such cases, the harmed groups can be determined entirely from the covariates.
>
> **R3: _'recruiting one treated and one untreated participant at each step' is unrealistic. Why not…have batch recruitment?_**
>
> Thank you for giving us the opportunity to clarify. This assumption is actually not necessary for CLASH, but rather for certain stopping tests that CLASH can be used with. For example, the mSPRT (Johari et al., 2017) requires this assumption, as it considers an “observation” at each time step to be the difference in outcomes between a treated and untreated unit. Note that this is not as strong an assumption as it may initially seem: if the data is i.i.d. (as we assume), then at any interim checkpoint, the data collected thus far can be re-ordered to ensure that there is a treated and untreated observation at every step. That said, it does complicate estimation in cases where the treatment-control split is not 50-50.
>
> However, we emphasize that CLASH does not require this assumption if used with non-SPRT techniques. For example, CLASH with the O’Brien-Fleming test can be used with batch recruitment with imbalanced treatment and control groups. This does not affect our results: for example, Figs 2b and 2c in our global response (which focus primarily on outcome variance) do not assume one treated and untreated participant per time step. We demonstrate that CLASH is still able to perform well in these settings.
>
> **R3: _What happens if the standard deviation is larger or smaller? Is it important to know this standard deviation in practice?_**
>
> Figs 2b and 2c in the global response consider these scenarios. We find that CLASH and the Oracle are both affected by increasing variance (Fig 2b). Notably, CLASH outperforms the homogenous baseline for all settings. Needing to estimate the variance (instead of knowing it a priori) does have an effect on CLASH’s performance (Fig 2c); however, CLASH still outperforms the homogeneous baseline across all considered variances.
>
> **R3: _What happens when the dimensionality is 100 and 1000 in the simulation?_**
>
> Fig 2e in the global response considers a high-dimensional setting with 100 and 500 covariates. We find that CLASH is still able to perform well with 100 covariates, outperforming the homogeneous baseline for medium and large effect sizes. With 500 covariates, CLASH only stops the experiment slightly more often than the homogeneous baseline for large effects. This result thus illustrates CLASH’s limitations in dealing with very high-dimensional covariate sets. We note, however, that 500 covariates is a fairly extreme setting for an experiment with 4,000 participants. In practice, even if 500+ covariates are available, practitioners would be able to use their domain expertise or statistical methods (e.g., LASSO) to specify a subset of covariates on which to assess harm. We have updated our limitations section to reflect this discussion, and recommend feature selection before running CLASH in very high-dimensional settings.
>
> **R3: _'We only sample from Region 1'... Why not sample uniformly at random…?_**
>
> This is a great question – in fact, we do present results from uniform sampling in Fig S14 in App H of the submitted paper. CLASH is able to outperform the homogeneous approach in this setting, though only for larger experiments (80k+ participants). However, CLASH’s main focus is on experiments in which a minority group of participants is harmed. If we were to uniformly sample all regions, the harmed group (Regions 1-4) would form the majority (60% of all participants); we thus only sample from Region 1 to illustrate our main contribution.

---

> > ### Comment · Reviewer_K2p1 · 2023-08-10
> > **Good response**
> >
> > Thank you for the complete response. I will raise my score to a 7.

---

### Official Review · Reviewer_af9S · 2023-07-20

**Soundness:** 3 good
**Presentation:** 3 good
**Contribution:** 4 excellent
**Rating:** 7
**Confidence:** 4

**Summary:**

This paper proposes a method to decide when to stop a clinical trial or an A/B study in cases where the treatment/intervention only causes harm to a minority group of participants, in which traditional methods can fail to detect the need for stopping an experiment. The paper includes a thorough theoretical analysis of the method, which suggests that it has desirable properties (stopping when a treatment is harmful and not stopping when it is not harmful) for large enough n. It also includes extensive simulations demonstrating that the method outperforms established methods in most cases and performs well in a real-world application.

**Strengths:**

- The paper is well written and easy to follow.
- The research question is interesting and important also from a fair AI perspective as this research protects minority groups.
- The paper includes a very extensive analysis of the problem, including theory, simulations and an empirical application.


**Weaknesses:**

- The simulations could include a wider range of sample sizes, i.e. sample sizes of 50-200 participants which are common in clinical trials.

**Questions:**

First of all, I want to say that this paper was a pleasure to read. I was impressed by the thoroughness and extend of the analyses. I only have a few suggestions. I hope you will find them helpful and constructive.

**Major points**
- I would like to see how the method holds up for much smaller sample sizes in simulations and on the empirical data, since there are many clinical trials with sample sizes around 50, 100 or 200 participants. Even if the method does not work well for this, it would be an important information to have so that practitioners do not apply this method when it’s not appropriate.
- On p. 3, you write “For example, consider a situation in which there are two equally sized groups with equal but opposite treatment effects, that is, p(G = 0) = p(G = 1) and τ (0) = −τ (1). The ATE is zero, and so any stopping test with H0 : ATE ≤ 0 is designed to continue to completion at least (1 − α)% of the time.” However, later you focus on Gaussian outcomes. Can your method handle the first scenario as well (i.e., bimodal distributions)? Or could it be extended to Gaussian mixtures for example?

- In the simulations shown in the supplement (Figure S6), there are certain scenarios for which the homogenous baseline outperforms CLASH (e.g., panel A: Bayesian Estimation MaxSPRIT, no treatment effect on the majority and small harmful effects or panel B: MaxSprit for small effects). Can you speculate on why this is the case? I do not think it is a problem if your method is not the best in every scenario, but it is important to outline and understand the conditions under which it is outperformed by other methods.

**Minor points**
- How would this method be used to decide for whom a trial should be stopped early in practice? Would this be based on the probability of being harmed, does this require a threshold? Perhaps you can elaborate more on this.
- Out of curiosity, could this method be extended to multiple groups that show different harm (e.g., a strongly and a weakly harmed group)?


**Limitations:**

The authors adequately discuss the limitations of their work. Depending on the performance for much smaller n in simulations it might be worthwhile expanded on the required sample size as well.

---

> ### Author Rebuttal · Authors · 2023-08-09
>
> Many thanks for your detailed review and positive feedback: we’re glad you enjoyed reading our paper and appreciate your encouragement. We address each of your constructive suggestions below, which we believe have further strengthened the paper.
>
> **R2: _The simulations could include a wider range of sample sizes, i.e. sample sizes of 50-200 participants which are common in clinical trials…I would like to see how the method holds up for much smaller sample sizes in simulations and on the empirical data, since there are many clinical trials with sample sizes around 50, 100 or 200 participants. Even if the method does not work well for this, it would be important information to have so that practitioners do not apply this method when it’s not appropriate._**
>
> Thank you for this important suggestion. Fig 2d in our global response considers an experiment with smaller sample sizes (N=200, 400, 1000). Overall, we find that CLASH can be effective in small samples: it outperforms the homogeneous baseline with as few as 400 participants, and with 1000 participants, the performance gap is wide. However, N=200 is a challenging scenario: neither CLASH nor the homogeneous baseline is able to stop experiments with such few participants. We have included this setting as a limitation in our revised paper.
>
> **R2: _On p. 3, you write “For example, consider a situation in which there are two equally sized groups with equal but opposite treatment effects, that is, p(G = 0) = p(G = 1) and τ (0) = −τ (1). The ATE is zero, and so any stopping test with H0 : ATE ≤ 0 is designed to continue to completion at least (1 − α)% of the time.” However, later you focus on Gaussian outcomes. Can your method handle the first scenario as well (i.e., bimodal distributions)? Or could it be extended to Gaussian mixtures for example?_**
>
> Apologies for the confusion on this point. Our simulation experiments do in fact consider this setting: the outcomes are Gaussian within each group (i.e., majority and minority), not the population as a whole. Thus, viewed over the whole population, the outcomes are generated from a mixture of two Gaussians. We have clarified this point in the revised paper.
>
> **R2: _In the simulations shown in the supplement (Figure S6), there are certain scenarios for which the homogenous baseline outperforms CLASH (e.g., panel A: Bayesian Estimation MaxSPRIT, no treatment effect on the majority and small harmful effects or panel B: MaxSprit for small effects). Can you speculate on why this is the case? I do not think it is a problem if your method is not the best in every scenario, but it is important to outline and understand the conditions under which it is outperformed by other methods._**
>
> Thank you for giving us an opportunity to address this point. In general, CLASH performs better when it is easy to differentiate between the harmed and unharmed groups from the covariates and outcomes. The more similar the effects on the minority and majority groups, the harder this task is; CLASH thus performs best when |majority effect - minority effect| is large. The homogeneous baseline is the exact opposite: it performs better when the whole population ATE is more similar to the effect on the minority group. Thus, the case in which the treatment has no effect on the majority but a small harmful effect on a large minority group (25% or 50%) is a near-ideal situation for the homogeneous baseline, but a more difficult situation for CLASH. However, it is worth noting that the difference in stopping probability between CLASH and the homogeneous baseline is relatively small, even in this difficult situation. We have added this discussion to the revised paper to provide more insight for practitioners.
>
> **R2: _How would this method be used to decide for whom a trial should be stopped early in practice? Would this be based on the probability of being harmed, does this require a threshold? Perhaps you can elaborate more on this._**
>
> Thank you for raising this important point. We have addressed this in point (2) of our global response. In short, the decision will heavily depend on domain specifics, but CLASH weights can provide heuristics that can help inform practitioner decisions.
>
> **R2: _Out of curiosity, could this method be extended to multiple groups that show different harm (e.g., a strongly and a weakly harmed group)?_**
>
> Yes, CLASH can be leveraged in this scenario: see Fig 1 in our global response for a demonstration.

---

> > ### Comment · Reviewer_af9S · 2023-08-11
> > **All questions answered and thank you**
> >
> > Thank you very much for addressing my concerns and including these additional experiments. I stand by my first score highlighting that this is a strong paper. I am certain that this work will be a great contribution to the conference. I am looking forward to reading more about your work and wish you a great conference.

---

### Official Review · Reviewer_wJ1C · 2023-07-24

**Soundness:** 4 excellent
**Presentation:** 4 excellent
**Contribution:** 3 good
**Rating:** 7
**Confidence:** 4

**Summary:**

A two-stage approach CLASH was proposed to determine the early stopping time of a randomized experiment when the treatment is harmful to a subset of the population. The indicators of the harmed groups are estimated by causal machine learning methods in stage 1, and the early stopping time is determined using the weighted test statistic in stage 2. Theoretical properties of the proposed method were established. Simulation studies were conducted to show the existing homogenous method's failure and CLASH's success. Finally, an illustration of the proposed methods was presented by analyzing real data from a digital experiment. Overall, the proposed methodology is useful, and the paper is well-written.

**Strengths:**

The method is solid, clearly described, and useful in different areas. With the accommodation of both Gaussian and time-to-event experimental outcomes and considering the limited prior work on this specific area, the proposed method can make a good impact.  The results of the simulation studies and real data analysis are convincing and support their conclusions.

**Weaknesses:**

The simulation settings seem relatively simple and difficult to interpret: only several binary covariates were included, and there was only 1 harmed group. It would be interesting to see some simulation results under settings closer to the real data analysis (multi-level categorical covariates and multiple harmed groups). Including several sentences to interpret the simulation settings would also be helpful.

**Questions:**

See Weaknesses.

**Limitations:**

The author provided a paragraph to discuss the limitations.

---

> ### Author Rebuttal · Authors · 2023-08-09
>
> Thank you for your detailed review and positive feedback: we’re glad you found our work useful and impactful. We appreciate your constructive comments, which we address below and in our revised paper.
>
> **R1: _The simulation settings seem relatively simple and difficult to interpret: only several binary covariates were included, and there was only 1 harmed group. It would be interesting to see some simulation results under settings closer to the real data analysis (multi-level categorical covariates and multiple harmed groups). Including several sentences to interpret the simulation settings would also be helpful._**
>
> In our global response, we summarize results from additional simulation experiments that evaluate CLASH in settings closer to real-data analysis, per your suggestions. Specifically, Fig. 1 presents results with multiple harmed groups, while Fig 2e presents results with 100 and 500 binary covariates (similar to many multi-level categorical covariates). We find that CLASH outperforms the homogeneous baseline in these more realistic simulation settings, and hope that this finding encourages use of CLASH in real-world experiments. We include these new results, as well as more detail on interpreting these simulation settings, in Section 4 of the revised paper.

---

> > ### Comment · Reviewer_wJ1C · 2023-08-10
> >
> > Thank you for your response. I have read the other reviews and rebuttals. I will keep my original score and agree to accept this paper.

---

### Author Rebuttal · Authors · 2023-08-09

We thank the reviewers for their positive comments and constructive feedback. In this global response, we focus on two themes raised by multiple reviewers: (1) additional experiments, and (2) the decision to stop only on the harmed group. We separately provide responses to individual reviewers.

### (1) Additional Experiments
The attached pdf contains figures showing CLASH’s performance in reviewer-suggested experiments. All figures use Gaussian outcomes and the O’Brien-Fleming stopping test; more experiments will be added to the paper supplement.

**Figure 1**: CLASH’s performance for trial populations with >2 groups. CLASH performs well: it stops more frequently than the homogeneous baseline across a range of effect sizes and as often as the Oracle for larger effects. We consider two settings:

a. Three groups of unequal size (group size and effect size, respectively, in parentheses): one weakly benefitted (87.5%,  -0.1), one strongly harmed (6.25%, x-axis), and one weakly harmed (6.25%, x/2).

b. Four equally sized groups (effect sizes in parentheses): strongly benefitted (-x), weakly benefitted (-x/2), weakly harmed (x/2), and strongly harmed (x).

**Figure 2**: Specific reviewer-suggested experiment settings. In all figures, the treatment harms the minority group and weakly benefits the majority (effect size: -0.1). Unless specified, there are 5 covariates and minority group size is 12.5%.

a. Stochastic group membership: covariates map stochastically to the benefited and harmed groups. We construct a 25% minority group deterministically from covariates as before, but randomly assign p% of this group to be harmed (the remainder is benefitted). CLASH outperforms the homogeneous baseline both when p=0.5 and 0.75.

b. Smaller/larger outcome variance: smaller / larger variance in the observed outcomes. CLASH outperforms the homogeneous baseline in all settings. CLASH and the Oracle both perform better with lower variance.

c. Estimated outcome variance: variance is not known, but rather estimated from the observed outcomes. Estimating the variance has an effect on CLASH’s performance, but CLASH still outperforms the homogeneous baseline.

d. Small sample sizes: small sample sizes (N = 200, 400, 1000). CLASH outperforms the homogenous baseline by a wide margin in experiments with moderately small samples (N=1000). While the gap between CLASH and the homogenous baseline decreases as sample size decreases, CLASH still outperforms the baseline with as few as 400 participants. However, with very small samples (N=200), neither CLASH nor the homogeneous baseline stops the experiment. We have included this setting as a limitation in our revised paper.

e. Large number of covariates: high-dimensional covariates (d=100, 500). CLASH is robust to increasing dimensionality to a point, outperforming the homogeneous baseline even with 100 covariates. The extreme case with 500 covariates is more challenging: here, practitioners may need to perform feature selection before running CLASH. We discuss this in our revised limitations section.

### (2) Decision to Stop Only on Harmed Group
We now focus on what investigators can do once CLASH indicates that an experiment should be stopped. We have added this discussion to the Methods section of the revised paper.

*Stopping Decision*. When CLASH indicates a group is being harmed, investigators should make choices based on domain expertise. If the nature of harm is serious (e.g., mortality) they may decide to stop the experiment for all participants. For milder harms (e.g., crashes in an A/B test), they may decide to stop the experiment only for the harmed group. If the identified harm is much less consequential than the potential benefit (e.g., harm of increased headaches vs. benefit of curing cancer), they may decide to not stop the experiment at all. The specific choice made will depend heavily on the ethical and financial aspects of the experiment, as well as a thorough review of the interim data. CLASH is not intended to make this decision for investigators; however, it helps investigators realize that a group is being harmed and thus a stopping decision needs to be made.

*Tree-based heuristic for identifying the harmed group*: If investigators choose to stop the experiment only for harmed participants, they must first identify the harmed group. The distribution of CLASH weights at stopping time can help in this task: groups with estimated participant weights close to 1 are likely to be harmed. Fig S13 in Appendix H illustrates this in our empirical application: the estimated weights in Regions 1 and 2 are both close to 1, indicating that these are the groups on which to stop. With few covariates, investigators can manually inspect the weight distribution for each covariate combination to identify the harmed group. With many covariates, investigators can use a simple heuristic: a regression decision tree on the estimated CLASH weights can find the covariate values for which the weights are the largest. Limiting the depth of this tree can ensure that the identified group is actionable (i.e., it is possible to stop the experiment on it) and of non-trivial size.

There are alternative approaches to this harmed group identification task; for example, practitioners can use subgroup identification methods on the raw outcomes (e.g. [1]). We are agnostic to the choice of method: investigators can pick the approach most appropriate for their domain.

*Treatment effect estimation*. Stopping the experiment in only one group can affect inference at the end of the experiment, as the treatment is no longer randomly assigned across covariates. To estimate the ATE over the entire population, practitioners should use inverse propensity weights to correct for the induced selection bias. We illustrate this correction in our response to reviewer K2p1 below, and in our revised manuscript.

[1] Zhang, et al, DOI: 10.21037/atm.2018.03.07

---

### Decision · Program_Chairs · 2023-09-21

**Decision:**

Accept (spotlight)

**Comment:**

This well-written paper has been assessed by 6 knowledgeable reviewers. They largely agree on accepting it: four gave straight accept, and two weak accept score. The authors provided thorough rebuttals that effectively addressed key concerns raised by the reviewers. The presented work is original and it addresses somewhat sparsely studied yet practically important problem. Considering all the above, I have no hesitation in recommending acceptance of this work, and if there is space, offering it a spot in a spotlight session.